# Hierarchical Epsilon-Net Graphs: Time Guarantees for HNSW in Approximate Nearest Neighbor Search

## Abstract

Hierarchical graph-based algorithms such as HNSW achieve state-of-the-art performance for Approximate Nearest Neighbor (ANN) search in practice, but they often lack theoretical guarantees on query time or recall due to their heavy use of randomized heuristic constructions. In contrast, existing theoretically grounded structures are typically difficult to implement and struggle to scale in real-world scenarios. We introduce a property of hierarchical graphs called Hierarchical $\varepsilon$-Net Navigation (HENN), grounded in $\varepsilon$-net theory from computational geometry. This framework allows us to establish time bounds for ANN search on graphs that satisfy the HENN property. The design of HENN is agnostic to the underlying proximity graph used at each layer, treating it as a black box. We further show that HNSW satisfies the HENN property with high probability, enabling us to derive formal time guarantees for HNSW. Direct construction of a HENN graph requires finding $\varepsilon$-nets. Existing methods for finding $\varepsilon$-nets are either probabilistic or, when deterministic, become impractical in high dimensions. To address this, we propose a budget-aware practical algorithm for building $\varepsilon$-nets, under a user-specified preprocessing time budget. Empirical evaluations confirm our theoretical guarantees for both HENN and HNSW, and demonstrate the effectiveness of the proposed budget-aware algorithm for constructing HENN and, more generally, $\varepsilon$-nets. This flexibility allows practitioners to select a method that best fits their specific use case.

## 1 Introduction

The Approximate Nearest Neighbor (ANN) problem involves retrieving the $k$ closest points to a given query point $q$ in a $d$-dimensional metric space. This problem is foundational in database systems, machine learning, information retrieval, and computer vision, and has seen growing importance in large language models (LLMs) [29], particularly in retrieval-augmented generation (RAG) pipelines, where relevant documents must be retrieved efficiently from large corpora [13; 34]. More generally, any vector database system implements some form of ANN search to enable efficient vector similarity queries [48; 19; 3]. For a comprehensive overview of additional applications, we refer the reader to the following surveys [54; 35; 37].

Several classes of algorithms have been developed for ANN. Hash-based approaches (e.g., Locality-Sensitive Hashing [21; 24; 8]) offer theoretical guarantees on retrieval quality but often struggle in practical settings [49]. Quantization-based methods cluster data and search among representative centroids, yielding speedups at the cost of approximation error [27; 16; 47]. Graph-based approaches [40; 26; 54], particularly hierarchical variants [40; 38; 42], have gained attention due to their strong empirical performance and scalability. These methods build graphs over the dataset and perform greedy traversal to locate approximate neighbors quickly.[1]

Among them, the Hierarchical Navigable Small World (HNSW) [40] graph is widely used in practice. HNSW organizes data into multiple layers by assigning each point to a randomly chosen level and constructs navigable small-world graphs at each layer. While HNSW is widely used in many existing tools and is easy to implement, it lacks formal guarantees on query time, and its worst-case

---

[1]Related work is further discussed in Appendix A.

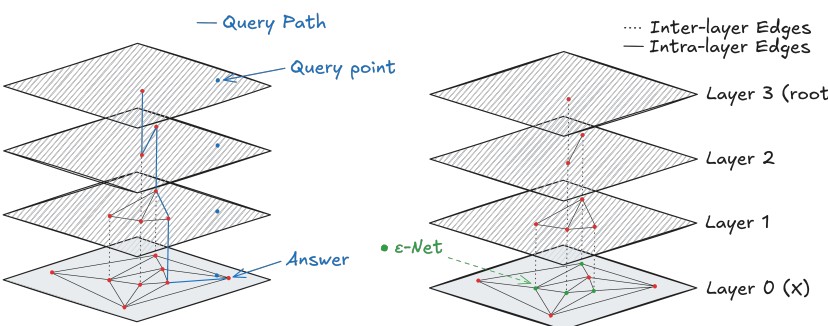

Figure 1: A simple representation of the Hierarchical $\varepsilon$-net Navigation Graph (right) with an example of answering a query using this structure (left). Layers are numbered bottom-up, with layer $0$ being the point set $X$ and the last layer (layer $L$) called the root.

complexity is shown to be *linear to dataset size* in *adversarial* settings [25; 50]. In contrast, earlier theoretically grounded hierarchical structures like Cover Trees [5; 31], provide logarithmic query time guarantees by constructing hierarchies using $r$-nets.[2] However, these structures are relatively *difficult to implement* and often do not scale well to real datasets, limiting their practical adoption.

In this work, we introduce a general class of hierarchical graphs for ANN search, grounded in $\varepsilon$-net theory from computational geometry [23]. In these structures, each layer forms an $\varepsilon$-net of the preceding one, while the choice of proximity graph at each layer remains agnostic, allowing the use of any suitable graph. Our framework establishes time guarantees for such indices as a function of the properties of the underlying proximity graph. Notably, we show that HNSW is a special case of HENN with high probability, corresponding to the choice of NSW as the base proximity graph. As a crucial building block, we further propose budget-aware practical algorithms for computing $\varepsilon$-nets on large datasets, enabling the scalable construction of these hierarchical graphs.

Our contributions can be summarized as follows:

- We introduce a general property for hierarchical indices in ANN search, called *Hierarchical $\varepsilon$-Net Navigation (HENN)*, where each layer is an $\varepsilon$-net of the preceding one. This framework is agnostic to the choice of proximity graph and can be combined with common graphs such as kNN, NSW [40], NSG [14], RNG [51], etc.
- We establish *probabilistic* query-time bounds for HENN graphs, which are logarithmic in the dataset size and parameterized by the properties of the underlying proximity graph.
- We show that HNSW is a HENN with high probability, when NSW is used as the proximity graph, thereby yielding formal probabilistic time guarantees for HNSW.
- A key component in building HENN graphs is computing $\varepsilon$-nets. Existing randomized algorithms [23] provide $\varepsilon$-nets only with a fixed success probability, while deterministic methods based on discrepancy theory [6] are impractical in high dimensions. We design a budget-aware algorithm that allows the user to specify a preprocessing budget. More preprocessing time increases the probability of successfully constructing an $\varepsilon$-net.

Our empirical evaluation confirms the theoretical time bounds for HENN and HNSW as a special case of HENN, as well as for other HENN-based structures. We show that these bounds hold with high probability (greater than $0.99$) in practical scenarios, and further demonstrate the flexibility of HENN by integrating it with different proximity graphs and comparing their performance. In addition, we illustrate how our budget-aware algorithm enables the construction of compressed indices: by allocating more preprocessing time, one can reduce index size while maintaining recall, highlighting a clear tradeoff between preprocessing cost and query performance. Finally, we implement HENN as a new index in the popular Faiss [12] library and compare it against state-of-the-art baselines, showing its equivalence to HNSW and empirically validating our theoretical claims.

**Paper Organization**. The paper is organized as follows. We begin by providing formal definitions and necessary background on $\varepsilon$-net theory and proximity graphs (Section 2). In Section 3, we introduce the HENN structure. The subsequent sections establish theoretical bounds (Section 4) and show that HNSW can be viewed as a HENN graph (Section 5). We then discuss practical aspects of

---

[2]Here, $r$-nets differ from $\varepsilon$-nets as defined in computational geometry. By $r$-net, we refer to a subset of points that ensures every point in the space is within distance $r$ of some net point. In contrast, $\varepsilon$-nets refer to subsets that intersect all "heavy" ranges, containing more than an $\varepsilon$ fraction of the total volume or weight.

constructing HENN graphs, including algorithms for computing $\varepsilon$-nets briefly in Section 6 (detailed in Appendix B), followed by our experimental evaluation in Section 7. The appendix contains related work, additional background on $\varepsilon$-nets and proximity graphs, results for parallel and dynamic settings, limitations, and further experimental details.

## 2 BACKGROUND AND DEFINITIONS

In this section, we introduce the concepts and notations that will be used throughout the paper.

**Data Model**. Let $X = \{x_i\}_{i=1}^n$ denote a set of $n$ points, where each $x_i$ is a $d$-dimensional vector in $\mathbb{R}^d$. We define a distance function $\mathbf{d} : \mathbb{R}^d \times \mathbb{R}^d \to \mathbb{R}$ over this space, resulting in the metric space $(X, \mathbf{d})$. For example, the $\ell_p$-norm between two points $x_i$ and $x_j$ is defined as

$$\mathbf{d}(x_i, x_j) = \|x_i - x_j\|_p = \left( \sum_{k=1}^d |x_i[k] - x_j[k]|^p \right)^{\frac{1}{p}}.$$

Unless otherwise stated, we use the $\ell_2$-norm in our examples and analysis. However, the results extend naturally to any metric space, yielding a bounded VC-dimension for the range families introduced later.[3]

**VC-dimension**. Let $\mathcal{R}$ be a family of ranges defined over $X$, such as balls, axis-aligned rectangles, or half-spaces in $\mathbb{R}^d$. The VC-dimension of $(X, \mathcal{R})$, denoted $\delta$, is the largest integer $m$ for which there exists a subset $S \subseteq X$ of size $m$ that is *shattered* by $\mathcal{R}$ [52; 20].

$\varepsilon$-**net**. Let $(X, \mathcal{R})$ be a range space with bounded VC-dimension $\delta$. A subset $\mathcal{N} \subseteq X$ is called an $\varepsilon$-*net* of $(X, \mathcal{R})$ if, for every range $R \in \mathcal{R}$ with $|R \cap X| \geq \varepsilon |X|$, we get $\mathcal{N} \cap R \neq \varnothing$. In other words, $\mathcal{N}$ intersects every "heavy" range; any range containing at least an $\varepsilon$-fraction of the points in $X$. We will often refer to an $\varepsilon$-net of $(X, \mathcal{R})$ simply as an $\varepsilon$-net of $X$ when the range family is clear from the context.

We make use of the following well-known result [23; 20].

**Theorem 1** *Let $(X, \mathcal{R})$ be a range space with VC-dimension $\delta$. If a random sample of size $m_\varepsilon$ is drawn with replacement, where*

$$m_\varepsilon \geq \max \left\{ \frac{4}{\varepsilon} \log \frac{4}{1 - \varphi}, \ \frac{8\delta}{\varepsilon} \log \frac{16}{\varepsilon} \right\},$$

*then the sample forms an $\varepsilon$-net with probability at least $\varphi$.*

Ignoring constant factors, Theorem 1 implies the existence of an $\varepsilon$-net of size $O\left(\frac{\delta}{\varepsilon} \log \frac{1}{\varepsilon}\right)$ for range spaces with VC-dimension $\delta$.[4] The theorem also provides a *randomized sampling* method for constructing $\varepsilon$-nets. Alternatively, $\varepsilon$-nets with the same size can be constructed *deterministically* using techniques from discrepancy theory [6]. More details on $\varepsilon$-net construction is provided in Section 6.

**Problem Setting**. Our objective is to preprocess $X$ and construct a data structure that enables answering the nearest-neighbor queries. Given a query point $q \in \mathbb{R}^d$ and a point set $X$, a $k$-nearest neighbor ($k$-NN) query aims to find the $k$ closest points $x_i \in X$ to $q$; i.e., $O_q = k\text{-}\min_{x_i \in X} \mathbf{d}(x_i, q)$. The Approximate Nearest Neighbor (ANN) is a relaxed version of this problem where the goal is to find a set $A_q \subset X$ of size $k$ with a high *recall rate*: the probability that each returned point in $A_q$ belongs to $O_q$, formally: $Recall@k = \frac{|A_q \cap O_q|}{k}$.

**Proximity Graph (PG)**. Given a point set $X$, a proximity graph is a graph whose vertices correspond to the points in $X$, and whose edges connect pairs of points according to a proximity criterion in the underlying space. The goal is to preserve the neighborhood structure of the data so that ANN

---

[3]Throughout our guarantees and proofs, we follow the convention in computational geometry [1] of removing the dimension $d$ as a multiplicative factor in big-oh notations. However, we explicitly retain $d$ when it appears in the exponent.

[4]Assuming a constant failure probability $\varphi$.

queries can be answered by navigating the graph. Several types of proximity graphs have been studied in the literature, including the Delaunay Triangulation (DT) [32], Navigable Small World graphs (NSW) [41], $k$-nearest neighbor graphs ($k$-NN), Relative Neighborhood Graphs (RNG) [51], and Navigable Spreading-out Graphs (NSG) [14]. A more detailed discussion of these graph types is provided in Appendix C.

## 3  HIERARCHICAL $\varepsilon$-NET NAVIGATION GRAPH (HENN)

In this section, we introduce a property for the hierarchical navigation graphs that guarantees their ANN query answering time.

**Definition 1 (Hierarchical $\varepsilon$-net Navigation Graph (HENN))** *A multi-layer graph built on top of the point set $X$ is a HENN graph (has HENN property) if it satisfies the following criteria:*

1. **Nodes:** *Each node of the graph represents a point in $X$.*
2. *$\varepsilon$-**net Hierarchy:** Each layer $\mathcal{L}_i$, for $0 \leq i \leq L$, is an $\varepsilon$-net of the preceding layer, with $\mathcal{L}_0 = X$. The $\varepsilon$-net is defined with respect to the ring ranges and for a specific value of $\varepsilon$, both introduced later (Equation 1).*
3. **PG (Intra-layer edges):** *The nodes within each layer are connected with* Intra-layer edges *that construct a* proximity graph *(PG) to answer the ANN* only *inside this layer. Any PG can be integrated into HENN.*
4. **Inter-layer edges:** *Each pair of layers $\mathcal{L}_i$ and $\mathcal{L}_{i+1}$, $0 \leq i < L$, are connected with Inter-layer edges. There is an edge between the nodes $v \in \mathcal{L}_i$ and $u \in \mathcal{L}_{i+1}$, if and only if $v$ and $u$ represent the same point in the point set $X$.*

We call a graph that satisfies the HENN property *a HENN graph*. Figure 1 (right) shows a simple example of a HENN graph, in which each layer contains half as many points as the layer below it. We now describe the HENN structure in detail, focusing on the high-level structure, as well as each individual component.

$\varepsilon$-**net Hierarchy**. Each layer $\mathcal{L}_i$ in a HENN graph is an $\varepsilon$-net, with a choice of $\varepsilon$ that results in the optimum query time (Equation 1). This $\varepsilon$-net is defined on the range space $(X, \mathcal{R})$, where $X$ denotes the input point set and $\mathcal{R}$ is the family of *ring* ranges defined below.

**Definition 2 (Ring Ranges)** *Given a set of points $X$, and a distance function $\mathbf{d}$, a **ring** $R \in \mathcal{R}$ is specified with a base point $p \in \mathbb{R}^d$ and two values $r_1 < r_2$. Any point in $X$ with distance within two values $r_1$ and $r_2$ from $p$ falls inside the ring. Formally,*

$$R \cap X = \{x \in X \mid r_1 \leq \mathbf{d}(x, p) \leq r_2\}$$

**Proposition 2** *The VC-dimension of the ring range space, $(X, \mathcal{R})$, is $\Theta(d)$.*

The correctness of Proposition 2 follows the fact that each ring range $R : \langle p, r_1, r_2 \rangle$ can be formulated by mixing the two ball ranges $R' : \mathbf{d}(x, p) \leq r_2$ and $R'' : \mathbf{d}(x, p) \leq r_1$ as $R = R' - R''$. Hence, due to the mixing property of range spaces [20], the VC-dim of the ring ranges is two times the VC-dim of the distance ranges, i.e., $\Theta(d)$. Remember that we assumed the distance $\mathbf{d}$ gives us a VC-dim of $\Theta(d)$ for balls.

Following Proposition 2 and Theorem 1, one can find an $\varepsilon$-net of size $m_\varepsilon = O(\frac{d}{\varepsilon} \log \frac{1}{\varepsilon})$, for the ring range space, for a given value $\varepsilon$. The details on the construction of $\varepsilon$-net is provided in Section 6. Hereafter, whenever we refer to an $\varepsilon$-net of a point set, we mean an $\varepsilon$-net with respect to the range space of rings defined on that point set.

**The Value of $\varepsilon$.**   To satisfy the HENN property, each layer $\mathcal{L}_{i+1}$ must form an $\varepsilon$-net of the preceding layer $\mathcal{L}_i$. We define a function $\mathcal{E} : \mathbb{N} \to (0, 1)$ that specifies the value of $\varepsilon$ for a layer of size $n$. In other words, $\mathcal{L}_{i+1}$ is an $\mathcal{E}(|\mathcal{L}_i|)$-net of $\mathcal{L}_i$. For the graph to be HENN, the required value of $\varepsilon$ is

$$\varepsilon = \mathcal{E}(|\mathcal{L}_i|) = \mathcal{E}(n) = \Theta\left(\frac{d \log n}{n}\right), \tag{1}$$

where $n = |\mathcal{L}_i|$.

In the next section, we show that this choice of $\varepsilon$ leads to optimal query time guarantees on the graph.

**The Construction of HENN**. Section 6 and Appendix B present the details of how to directly construct a HENN graph, including the methods for finding $\varepsilon$-nets. For a high-level perspective, we also provide pseudo-code outlining the construction procedure in Algorithm 1. Until then, we assume the graph is given and focus on establishing guarantees for it.

**Query Answering Using a HENN Graph**. During the query time, given a query point $q \in \mathbb{R}^d$, the goal is to find the approximate nearest neighbor of $q$ within $X$. We follow the *greedy search* algorithm on top of HENN graph: Starting from a random node in the root (layer $\mathcal{L}_L$), we find the nearest neighbor of $q$ within this layer by following a simple greedy search algorithm [41; 40]. After finding the nearest neighbor $v_i$ in layer $\mathcal{L}_i$, we continue this process, starting from $v_i$ in layer $\mathcal{L}_{i-1}$. We proceed until reaching the bottom (layer $\mathcal{L}_0$).[5] A pseudo-code of this algorithm is provided in Algorithm 5 in the Appendix, and a visual illustration is shown in Figure 1 (left). Answering $k$-Nearest Neighbors for $k > 1$ can be achieved by considering a set of candidates at each step in the greedy algorithm or running a Beam Search to consider multiple paths to the query [40; 26].

## 4 THEORETICAL ANALYSIS

In this section, we analyze the query-time complexity of the HENN structure with respect to the black-box choice of the underlying proximity graph (PG) and the value of $\varepsilon$ specified in Equation 1. We begin by introducing the preliminary concepts needed to establish the final time bound.

### 4.1 DEFINITIONS

Let $\mathsf{GS}_q(\mathcal{G}, s)$ denote the result of running the greedy search algorithm (Algorithm 5) on a proximity graph $\mathcal{G}$ for the query $q$, starting from node $s$. Let $\mathcal{G}_\mathcal{L}$ denote the proximity graph constructed at layer $\mathcal{L}$ of HENN. We now introduce the following definition, which captures a property of the specific proximity graph under consideration:

**Definition 3 (Recall Bound $\rho_\gamma$)** *Let $\mathcal{G}$ be a proximity graph defined on a point set $X$. The Recall Bound of $\mathcal{G}$ is the smallest integer $k$ such that, for all queries $q$, greedy search on $\mathcal{G}$ returns at least one point among the $k$-nearest neighbors of $q$ with probability at least $\gamma$. Formally,*

$$\rho_\gamma := \min\Big\{ k \ \Big| \ \Pr_{s,q}[\, \mathsf{GS}_q(\mathcal{G}, s) \in \mathsf{NN}_{k,X}(q) \,] \geq \gamma \Big\}.$$

*where $\mathsf{NN}_{k,X}(q)$ denotes the ground-truth $k$-nearest neighbors of $q$ in $X$, and the probability is taken over the queries and choices of the starting node.[6]*

This definition captures a weaker notion of accuracy for proximity graphs compared to the standard recall metric. While $\mathrm{Recall}@k$ measures the fraction of the true $k$ nearest neighbors retrieved, $\rho_\gamma$ identifies the smallest $k$ such that, with high probability, *at least one* of the true $k$ nearest neighbors is returned by the search algorithm. In our analysis, we fix a value of $\gamma$ and consider the corresponding $\rho_\gamma$. However, one can generalize analysis by varying $\gamma$, which yields to a full distribution of the recall bound. Additional details on how to interpret the recall bound are provided in Appendix K.

### 4.2 RUNNING TIME ANALYSIS

We begin with a simplified setting where HENN consists of only two layers: the base layer $\mathcal{L}_0 = X$ and the upper layer $\mathcal{L}_1$, which is an $\mathcal{E}(n)$-net of $X$, where $n = |X|$. A greedy search is initiated from an initial node in $\mathcal{L}_1$, proceeds until it reaches a *local minimum* in this layer, and then continues in the base layer. The following lemma provides an upper bound on the total number of steps taken after reaching the local minimum in $\mathcal{L}_1$:

**Lemma 3** *Let $\mathcal{L}_1$ and $\mathcal{L}_0$ be defined as above. Let $p = \mathsf{GS}(\mathcal{G}_{\mathcal{L}_1}, s)$ denote the result of running $\mathsf{GS}$ from an initial node $s \in \mathcal{L}_1$. Then, with probability at least $\gamma$, the number of points in $\mathcal{L}_0$ that are closer to $q$ than $p$ is $O(\rho_\gamma \cdot \varepsilon \cdot n)$ where $\varepsilon = \mathcal{E}(n)$.*

**Proof** *Sketch: This is a result of finding $\varepsilon$-nets on ring ranges, where it bounds the total number of points around $q$. The proof is provided in Appendix G.* □

---

[5]Note that this is the standard greedy algorithm used in the literature.
[6]Depending on the specific PG, this starting node can be uniformly random or even deterministic.

We now analyze the query running time of HENN and show that the choice of $\mathcal{E}$ in equation 1 yields the optimal runtime. Throughout this analysis, we assume access to an algorithm that, for a given $\varepsilon$, computes an $\varepsilon$-net of size $m_\varepsilon$ (see Theorem 1) with probability at least $\varphi$. A more detailed discussion of the construction algorithms is provided in Section 6.

**Theorem 4** *Let a HENN index be given, as defined in Definition 1, constructed using a proximity graph $\mathcal{G}$ at each layer, with recall bound $\rho_\gamma$ for some choice of $\gamma$. Assume further that each layer forms an $\varepsilon$-net of size $m_\varepsilon$ with probability at least $\varphi$, where $\varepsilon$ is chosen as in Equation 1: $\mathcal{E}(n') = \Theta\left(\frac{d \log n'}{n'}\right)$. Let $n = |X|$ be the size of the point set. Then, for the number of layers $L = \log n$, the query running time is $O(\rho_\gamma \pi d \log^2 n)$, which holds with probability at least $(\varphi\gamma)^{\log n}$.*

**Proof** *Sketch: This is the result of applying Lemma 3 inductively on all layers. See Appendix G for the proof.* $\square$

Theorem 4 also shows that the best choice of function $\mathcal{E}$ is $\mathcal{E}(n) = \frac{cd \log n}{n}$ for $c$ to be a constant.[7] This implies that for a layer $\mathcal{L}_i$ of size $n'$, the next layer $\mathcal{L}_{i+1}$ satisfies

$$|\mathcal{L}_{i+1}| = m_{\mathcal{E}(n')} = O\left(\frac{d}{\mathcal{E}(n')} \log \frac{1}{\mathcal{E}(n')}\right) \leq O\left(\frac{n'}{c}\right).$$

In other words, by choosing a sufficiently large constant $c$, the size of each layer decreases exponentially relative to the previous one.

For a choice of PG and a general $\gamma$ that results in a recall bound $\rho_\gamma$, the query time will be $O(\rho_\gamma d \log^2 n)$. By choosing different values of $\gamma$, we obtain a spectrum of query times with probability at least $(\varphi\gamma)^{\log n}$, which provides more information about the probability distribution of the query time. More details on deriving a distribution over query time is provided in Appendix **??**.

**Space Usage**. Since each layer decreases in size by at least a constant factor $c > 1$, the layer sizes form a geometric progression. Consequently, the total number of points stored across all layers is bounded by $\sum_{i=0}^{L} |\mathcal{L}_i| = \sum_{i=0}^{L} O\left(\frac{n}{c^i}\right) = O(n)$. Therefore, the overall space usage of a HENN graph is linear in $n$, under the assumption that each proximity graph requires a linear space to the size of its underlying dataset.

## 5 HNSW IS A HENN GRAPH

In this section, we present one of our main contributions: showing that HNSW is a HENN graph, with a high probability. Building on this connection, we derive probabilistic time guarantees for HNSW.

We begin with a simplified description of the HNSW [40]. During the preprocessing, points are inserted incrementally: for each new node, a maximum layer level is assigned at random, where the probability of being placed in higher layers decreases exponentially. Once the layers for the node are determined, the node is connected to its nearest neighbors among the inserted nodes at those layers, found using the greedy search procedure (Algorithm 5).

Equivalently, one can view HNSW as starting from the base layer that contains all points, and then recursively building higher layers by **randomly sampling** subsets of points, so that the size of each layer decreases exponentially with the level. At each layer, an NSW graph [41] is constructed on the sampled set. Examining the definition HENN,[8] We observe that HNSW has the HENN property (probabilistically), where the underlying proximity graph is chosen to be NSW and the $\varepsilon$-net construction is implemented via random sampling (see Theorem 1).

### 5.1 TIME COMPLEXITY ANALYSIS OF HNSW

Let $c > 1$ denote the constant parameter in the HNSW index construction that controls the rate of layer size reduction. Specifically, for every $i \geq 1$ we have $|\mathcal{L}_i| = \frac{|\mathcal{L}_{i-1}|}{c}$. We now turn to the following key question to analyze each layer of the HNSW graph: "BASED ON THEOREM 1, WHAT ARE THE VALUES OF $\varepsilon$ AND $\varphi$ THAT GUARANTEE A RANDOM SAMPLE OF SIZE $\frac{n}{c}$ FORMS AN $\varepsilon$-NET WITH PROBABILITY AT LEAST $\varphi$?"

---

[7]See the proof in Appendix G

[8]Also see Section 6 for an example of construction algorithm.

**Lemma 5** *For any layer $\mathcal{L}_i$ of size $n$ in the HNSW index, the next layer $\mathcal{L}_{i+1}$ is an $\varepsilon$-net of $\mathcal{L}_i$ with $\varepsilon = \Theta\left(\frac{d \log n}{n}\right)$, and with probability at least $\varphi = 1 - \Theta\left(\frac{\log n}{n}\right)$.*

**Proof** *Sketch: This is a result of applying the size in Theorem 1. See Appendix G for proof.* □

Combining Theorem 4 and Lemma 5, for HNSW we conclude that the function $\mathcal{E}$ in this graph satisfies: $\mathcal{E}(n) = \Theta\left(\frac{d \log n}{n}\right)$. Thus, HNSW implicitly uses the optimal choice of $\mathcal{E}$ for $\varepsilon$ values. Consequently, the query running time is $O(d \log^2 n)$, with a probability of at least

$$\left((1 - \tfrac{\log n}{n}) \cdot \gamma\right)^{\log n},$$

where the parameter $\gamma$ depends on the recall quality of the NSW proximity graph (which has a constant degree $\pi$).

**Success Probability in Practice**. Assuming a constant recall bound $\rho_\gamma$ for NSW, the above probability simplifies to $(1 - \frac{\log n}{n})^{\log n}$. For concreteness, we can evaluate the probability for different dataset sizes: (1) When the dataset size is $n = 10^3$, we have $\left(1 - \frac{\log 1000}{1000}\right)^{\log 1000} \approx 0.905$. (2) When the dataset size is $n = 10^6$, we obtain $\left(1 - \frac{\log 10^6}{10^6}\right)^{\log 10^6} \approx 0.9996$.

These calculations explain why, in practice, HNSW often exhibits logarithmic query times in most cases. Observing this in HNSW therefore also provides indirect evidence of the effectiveness of NSW, in terms of recall bound, as the underlying proximity graph.

## 6 HENN CONSTRUCTION

Table 1: Comparison of different methods for constructing $\varepsilon$-nets.

| Algorithms | Output Guarantee | Time Guarantee | Practical |
|---|---|---|---|
| Sampling-based [23] | Probabilistic | Fast | ✓ |
| Discrepancy-based [6] | Deterministic | Slow (exponential to $d$) | ✗ |
| Sketch-and-Merge [6] | Deterministic | Near-linear to $n$ | ✗ |
| *Budget-Aware* (ours) | Probabilistic (Budget-based) | Fast (budget-based) | ✓ |

A detailed discussion of the preprocessing phase of HENN, along with an algorithm for constructing HENN graphs directly, is deferred to Appendix B. The core element of this algorithm is the computation of $\varepsilon$-nets. As our final contribution, we introduce and analyze a budget-aware algorithm for building $\varepsilon$-nets, also detailed in Appendix B. A summary of existing approaches and our proposed method is provided in Table 1.

## 7 EXPERIMENTS

In this section, we empirically validate the theoretical results of the proposed HENN structure and assess the algorithms developed for its construction. Furthermore, we implement the general HENN graph within the widely used Faiss library [30; 12], providing it as a new index alongside the existing popular ANN indices. The code is publicly available at this anonymous repository. Additional experimental results are provided in Appendix I due to space constraints.

We organize the experiments into the following parts:

1. **Experiments Setup.** Description of datasets (both real and synthetic), baseline methods, evaluation metrics, and configuration details.
2. **Proximity Graph Integration.** Integration of different proximity graphs into the HENN framework and evaluation of their performance.
3. **Verification of Time Guarantees.** Empirical validation of the time bounds proved in Section 4.
4. **$\varepsilon$-net Construction.** Comparison of introduced algorithms for building $\varepsilon$-nets (Section 6) and their impact when integrated into HENN.
5. **Comparison with Other ANN Indices.** Evaluation of a standard implementation of HENN integrated with the HNSW index in the widely used FAISS library, and comparison against other indices available there, including LSH [8], IVF-PQ [27], and NSG [14].

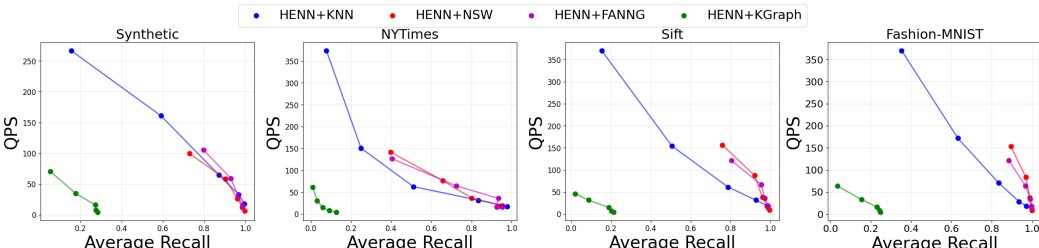

Figure 2: Comparison of HENN integrated with different proximity graphs. Points closer to the upper-right indicate better performance. QPS (queries per second) is the inverse of the average query time. the synthetic dataset contains 20k points with $d = 4$ following mixture of Gaussians.

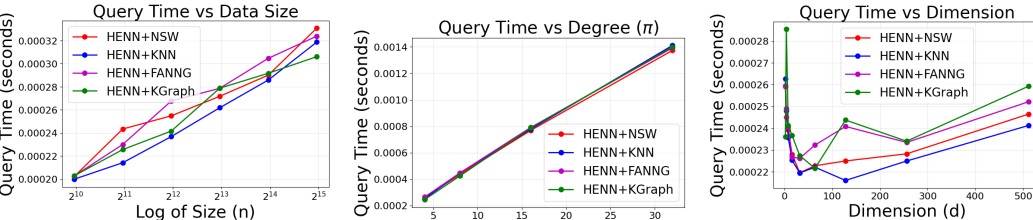

Figure 3: Effect of dataset size $n$ on the query time (GIST dataset).

Figure 4: Effect of PG degree on the query time (SIFT dataset).

Figure 5: Effect of dimension $d$ on the query time (synthetic uniform).

6. **Proximity Graph Comparison.** Analysis of recall bounds and performance trade-offs across different proximity graphs (provided in Appendix I).

**Experiments Setup**. We use standard benchmark datasets for ANN search [2], including SIFT-128 [28], GIST-960 [28], FASHION-MNIST-784 [56], and NYTIMES-256 [45], along with synthetic datasets (uniform distributions and mixtures of Gaussians). Both Euclidean ($\ell_2$-norm) and cosine (angular) distance metrics are considered. For proximity graphs, we evaluate several structures, including KNN, KGRAPH [11],[9] NSW [41], NSG [14], and FAANG [22]. Our methods are denoted as HENN+X, where $X$ specifies the underlying proximity graph used in the construction. Additional details on experimental setup can be found at the Appendix I.

**Proximity Graph Integration**. Figure 2 presents the performance of HENN when integrated with different proximity graphs across different datasets. The plots show the trade-off between query speed and recall as the exponential decay rate (the number of layers) in HENN are varied. Increasing the number of layers generally improves recall but slows down query processing, whereas fewer layers yield faster queries at the cost of weakening the hierarchical structure. Overall, HENN+NSW and HENN+FAANG achieve the best area under the curve (AUC).

**Verification of Time Guarantees**. We study the impact of several parameters on query time. Figure 3 illustrates the *effect of dataset size* $n$ on query time. For each dataset, we subsampled $n$ points and generated random queries within the space. Across all datasets, the query time grows logarithmically with $n$, consistent with Theorem 4. Similarly, Figure 15 (in Appendix I) reports the number of visited hops during greedy search with a similar trend. Similar experiments on more datasets is provided in Appendix I.

The effect of the *proximity graph degree* is shown in Figure 4, where we varied the degree across all layers. The results indicate that query time scales linearly with the graph degree $\pi$. Figure 5 examines the *effect of data dimensionality* $d$ on query time. Starting from $d = 2$, as $d$ increases, the search is more likely to get trapped in local minima (curse of dimensionality), which accelerates query by halting the greedy search early. However, once $d$ exceeds 64, the query time becomes dominated by the linear dependence on $d$. Similar results on the number of visited hops and other datasets are in Appendix I.

$\varepsilon$**-net Construction**. As discussed in Section 6 (and Appendix B), the budget-aware algorithm introduces a trade-off between the preprocessing budget and the success rate of the resulting $\varepsilon$-net. Figure 18 (in Appendix I) illustrates this trade-off: larger budgets $\mathcal{B}$ significantly increase the probability that the final set forms a valid $\varepsilon$-net. For example, on the FASHION-MNIST dataset,

---

[9]Here, KGRAPH refers to NN-descent applied on a $k$NN graph, following [11].

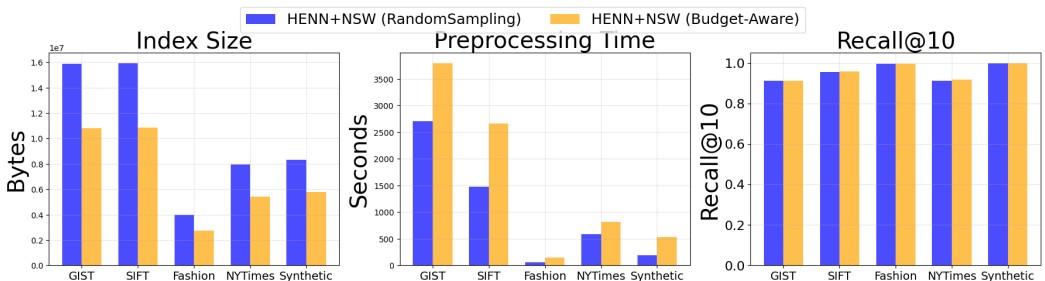

Figure 6: HENN index compression (with a cost of more preprocessing time) as an effect of using the Budget-Aware algorithm for finding $\varepsilon$-nets.

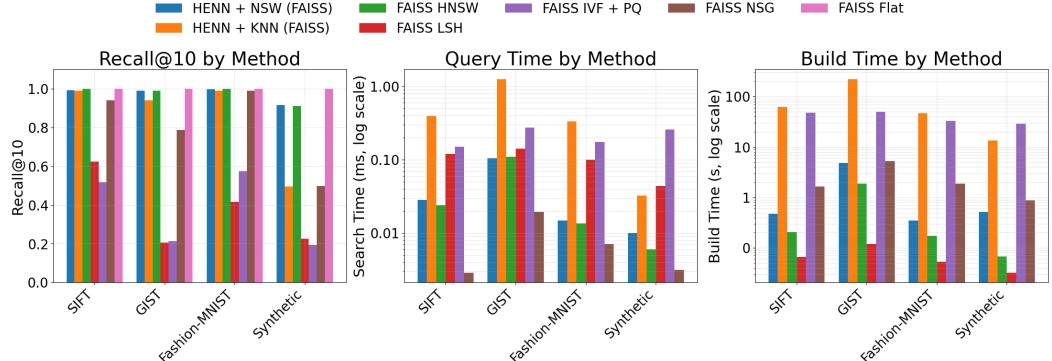

Figure 7: Comparison of HENN implementation in Faiss library with other indices.

allocating only 30% of the sampled points through finding unhit-sets raises the success rate from 0.7 to 1.0.

More importantly, Figure 6 demonstrates the impact of the budget-aware algorithm on the final HENN index. *Compressing* the index by selecting smaller subsets as $\varepsilon$-nets normally reduces the success rate ($\varphi$), but integrating the budget-aware strategy substantially changes this effect. As a result, HENN+BUDGETAWARE produces a more compact index with the same recall as HENN+NSW (or equivalently, HNSW). In other words, by spending additional preprocessing time, we obtain a smaller index, with a higher probability of each layer being an $\varepsilon$-net, resulting in no loss in recall.

**Comparison with Other ANN Indices**. We integrated HENN as a new index within the popular Faiss library and compared it against widely used ANN indices. Figure 7 presents this comparison. As expected, HENN+NSW and HNSW exhibit nearly *identical performance* in both recall and query time, consistent with our discussion in Section 5. The only noticeable difference is a slightly higher preprocessing time for HENN+NSW, which arises from employing the construction Algorithm 1 rather than the highly optimized incremental procedure used in HNSW.[10]

**Additional Experiments**. Further results on parameter variations, such as exponential decay, comparisons of recall bounds across different proximity graphs, index size, and preprocessing time are provided in Appendix I.

**Discussion and Limitations**. We defer a discussion on the parallelization, dynamic setting, and limitation of our work to the Appendix D.

## 8 CONCLUSION

We introduced HENN, a structural property for hierarchical graph-based indices in ANN search that unifies theoretical guarantees with practical efficiency. By organizing layers as $\varepsilon$-nets, HENN achieves provable polylogarithmic query time while retaining a simple and implementable design. We further provided a probabilistic analysis of HNSW, shedding light on the reasons behind its strong empirical performance. To support practical adoption, we developed a budget-aware algorithm for $\varepsilon$-net construction, allowing practitioners to balance preprocessing time against recall quality.

---

[10]Our objective in this study is not to provide a comprehensive comparison of HENN against all ANN baselines, but rather to demonstrate the equivalence between HENN+NSW and HNSW, since HENN represents a structural property rather than a specific graph.

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

APPENDIX

TABLE OF CONTENT

## A  RELATED WORK

In this section, we review the literature most relevant to our work.

**Hierarchical Methods for ANN**    A notable class of approaches for solving the ANN problem is based on hierarchical structures. One of the most widely used methods in this category is Hierarchical Navigable Small World (**HNSW**) [40], which constructs a multi-layered structure of navigable small-world graphs to enable efficient search. Interestingly, the core idea of hierarchical organization can be traced back to earlier work in computational geometry, including **Cover Trees** [5] and **Navigating Nets** [31].

Hierarchical Navigable Small World (HNSW) graphs [40] construct a multi-layered hierarchy of navigable small world (NSW) graphs. An NSW graph serves as an efficient approximation of the Delaunay graph [20], which is known to be an optimal structure for solving the approximate nearest neighbor (ANN) problem [41]. The Delaunay graph is closely related to the Voronoi diagram, which partitions the space into cells based on their proximity to the points in the dataset. Unlike the Delaunay graph, which requires explicit geometric computation, NSW graphs are built incrementally by inserting points and connecting them to their approximate nearest neighbors [41]. During insertion and querying, HNSW employs a greedy search algorithm (Algorithm 5) to navigate through the graph and locate nearby points. To improve scalability, HNSW organizes the data in a hierarchy where the number of nodes decreases exponentially across layers, resulting in $O(\log n)$ layers and a total space complexity of $O(n)$.

The Cover Tree [5] and Navigating Nets [31] are hierarchical data structures for nearest neighbor search in general metric spaces. Both achieve logarithmic query times by recursively organizing data into nested layers, cover trees through covering and separation invariants, and navigating nets via sequences of $r$-nets that approximate the dataset at multiple scales. While they offer strong theoretical guarantees and predictable performance, these methods are challenging to implement and scale poorly in practice, limiting their adoption in modern large-scale applications.

Beyond HNSW, several other hierarchical indices and heuristics have been proposed. None being generalized to HENN. For example, HVS organizes data into coarse Voronoi regions that are refined hierarchically, enabling layered navigation and accelerating search by progressively narrowing the candidate set [38]. HCNNG instead builds multiple hierarchical clusterings and merges them into a proximity graph, leveraging both global and local structure. By combining clustering with MST-based connectivity, HCNNG reduces construction overhead while maintaining competitive query performance [42].

Some methods integrate hierarchical tree structures with graph refinement, while large-scale libraries such as FAISS [30] adopt hierarchical inverted file strategies. In FAISS, coarse quantization partitions the dataset into clusters, which are then refined with product quantization (PQ) [27].

**Other Solutions for ANN**    Solutions to the nearest neighbor problem can be categorized along several dimensions. One common distinction is between classical methods, which primarily target the exact NN problem. Examples include k-d trees [4], ball trees [46], and Delaunay triangulations [21]. While effective in low-dimensional spaces, these methods typically fail to scale in high-dimensional settings. Another broad category includes quantization-based methods [27; 16; 47], which cluster the data and represent points by their assigned centroids (codewords), thereby approximating distances efficiently. Additionally, hashing-based methods such as Locality-Sensitive Hashing (LSH [8]) provide theoretical guarantees and have been widely used for high-dimensional ANN, and finally, the graph-based methods [41; 40; 50; 38]. For comprehensive overviews of these and other approaches, we refer the reader to the following surveys [48; 19; 3; 54; 35; 37].

Beyond graph topology, several lines of work improve ANN accuracy and efficiency through quantization. Guo, et al. [18] introduce direction-sensitive quantizers that better preserve angular similarity, enabling faster high-dimensional inference with competitive recall. Similarly, Rabitq [15] provides a quantization scheme with theoretical reconstruction-error guarantees designed specifically for ANN search, showing that quantization can substantially reduce memory while retaining accuracy. These works mainly focus on compressing the data vectors used for distance estimation, whereas our focus is on the structure and navigability of the search graph itself. Another complementary line is probabilistic search guidance [39], which proposes stochastic transition rules to escape local minima and improve robustness of greedy search. Such methods modify the search dynamics rather than the graph structure. The above systems are orthogonal and compatible with our theoretical contributions. Quantization-based methods can be applied inside any graph index to reduce vector memory footprint. Similarly, probabilistic routing modifies how the search moves on a fixed graph.

**Worst-Case Performance Analysis.**    Several works study the limitations of graph-based ANN indices. Indyk et al. [25] show that HNSW can require linear query time on adversarial datasets. Wang et al. [55] propose Steiner-hardness as a graph-native measure of query difficulty, capturing structural factors that influence cost and enabling the design of unbiased workloads. While these approaches characterize hard queries or highlight specific *failure cases on proximity graphs*, our analysis takes a complementary view by providing general bounds on query time for HENN as a function of the underlying proximity graph. Moreover, empirical results suggest that the worst-case recall bounds rarely manifest in practice, as they remain stable across real-world datasets.

**Comparison with Other Structural Analyses.**    Other recent works focus on improving or analyzing proximity graph constructions. Yang et al. [57] revisit construction strategies for indices like RNG and NSWG, optimizing pruning and edge selection to reduce build time without harming query performance. Diwan et al. [10] study the fundamental limits of navigable graphs, proving upper and lower bounds on node degree for efficient greedy routing. In contrast, our contribution is structural: we introduce HENN as a graph-agnostic framework based on $\varepsilon$-net layering, yielding provable polylogarithmic query-time guarantees for any underlying proximity graph.

DiskANN [26] shows that a navigable graph combined with SSD-aware prefetching enables billion-scale search with near–HNSW accuracy. Lin & Zhao [36] provide a critical examination of proximity-graph search, showing that well-constructed flat graphs can perform competitively and highlighting when greedy search succeeds or fails. Practical engineering optimizations also matter: Coleman et al. [7] demonstrate that simple node reordering can substantially reduce cache misses and latency. Most recently, Munyampirwa et al. [43] show that the performance of HNSW largely comes from a set of high-degree "hub" nodes, and that flattening the hierarchy yields similar recall and speed.

Although our theorems are stated for hierarchical structures, the same $\varepsilon$-net sampling idea can be applied to a single-layer graph: selecting an $\varepsilon$-net and adding long-range shortcut edges between its representatives mimics the effect of "hub" nodes. After following such a shortcut, our bounds control the remaining number of steps needed to reach the true neighbor. Thus, our framework not only explains hierarchical graphs like HNSW but also provides a structural justification for recent flat, hub-based designs.

# B HENN CONSTRUCTION

In this section, we first present an algorithm for directly constructing a HENN graph, followed by a discussion on building $\varepsilon$-nets, which serve as a key subroutine in the construction.

The preprocessing phase of building a HENN graph follows a recursive process provided in Algorithm 1. As an input, it receives a function $\mathcal{E} : \mathbb{N} \to (0, 1)$, that calculates the value of $\varepsilon$ as a function of input size $n$ (see Equation 1 for the best choice of this function).

It begins with the initial set of points being the entire point set, i.e., $\mathcal{L}_0 = X$, and constructs an $\varepsilon$-net over $X$, where $\varepsilon = \mathcal{E}(|X|)$. This forms the first layer, denoted $\mathcal{L}_1$. After finding the points in this layer, we follow a black-box approach for constructing a *proximity graph* within this layer and add the intra-layer edges accordingly (see Appendix C for more details).

---

**Algorithm 1** HENN Construction (Preprocess) Algorithm

---

**Require:** The set of points $X$, maximum number of layers $L$, and the function $\mathcal{E}$.
**Ensure:** The HENN graph $\mathcal{H}$.
  1: **function** BUILDHENN($X, L, \mathcal{E}$)
  2:     $\mathcal{L}_0 \leftarrow X$
  3:     **for** $i \le L$ **do**
  4:         $\varepsilon \leftarrow \mathcal{E}(|\mathcal{L}_{i-1}|)$
  5:         $\mathcal{L}_i \leftarrow BuildEpsNet(\mathcal{L}_{i-1}, \varepsilon)$            ▷ See Appendix B.1
  6:         Connect each node in $\mathcal{L}_i$ to the previous layer (Inter-layer edges).
  7:         Build a proximity graph on $\mathcal{L}_i$ (Intra-layer edges).       ▷ See Appendix C.
  8:     **Return** $\mathcal{H} = \{\mathcal{L}_0, \mathcal{L}_1, \cdots, \mathcal{L}_L\}$ and the edges.

---

Subsequently, the algorithm recursively builds each layer $\mathcal{L}_{i+1}$ as an $\mathcal{E}(|\mathcal{L}_i|)$-net of the previous layer $\mathcal{L}_i$ and adds the inter-layer edges. This process continues until a total of $L$ layers are constructed, where $L$ is a hyperparameter specifying the depth of the HENN graph. Even though this construction follows a sequential order in building layers, a discussion on how to parallelize this step is provided in Appendix F.

In the next sections, we present several algorithms for constructing $\varepsilon$-nets, which serve as a crucial subroutine in the preprocessing phase of HENN. Each algorithm is suitable for different settings, offering a range of trade-offs between speed, guarantees, and practicality. This allows a practitioner to select the method that best matches their requirements. A comparative summary of these algorithms is provided in Table 1.

## B.1 EXISTING $\varepsilon$-NET CONSTRUCTION ALGORITHMS

We begin with an overview of existing algorithms from the computational geometry literature. Then, as our third contribution, we introduce a new algorithm that balances preprocessing time and the success probability of $\varepsilon$-net construction, based on a user-specified budget.

**Sampling-based.** The *random sampling* algorithm (Theorem 1) offers a fast and practical way to construct $\varepsilon$-nets: it simply selects $m_\varepsilon$ random points from the dataset without enumerating ranges $\mathcal{R}$. However, the guarantee holds only with a certain probability, making this approach inherently non-deterministic. Also, repeating the process multiple times using a Las-Vegas algorithm is not practical (see Algorithm 2).

**Discrepancy-based.** When exact deterministic $\varepsilon$-nets are required, e.g., for robustness, algorithms from discrepancy theory [6] can be applied. Recursively partitioning the dataset yields a guaranteed $\varepsilon$-net in $O(n|\mathcal{R}|) = O(n^{\delta+1})$ time. The sketch-and-merge technique [6; 21] improves runtime to $O(\delta^{3\delta} \cdot \frac{1}{\varepsilon^2} \log(\frac{\delta}{\varepsilon}) \cdot n)$, though these methods remain impractical due to their exponential dependence on the dimension $d$ (see Algorithm 3).

The algorithm (Algorithm 3) works by iteratively *halving* the point-set $X$, until reaching the desired size of $c_0 \frac{\delta}{\varepsilon} \log \frac{\delta}{\varepsilon}$, where $c_0$ is a large enough constant.

---

**Algorithm 2** Building $\varepsilon$-net (Sampling-based Algorithm)

---

**Require:** The range space $(X, \mathcal{R})$, value of $\varepsilon$, failure probability $\varphi'$.
**Ensure:** The $\varepsilon$-net $\mathcal{A}$.
 1: **function** BUILDEPSNETSAMPLING($X, \varepsilon, \varphi'$)            ▷ Sampling-based algorithm
 2:     **repeat**
 3:        $m_\varepsilon \leftarrow$ calculate the size (Theorem 1)
 4:        $\mathcal{A} \leftarrow m_\varepsilon$ random samples with replacement from $X$.
 5:     **until** ISEPSNET($\mathcal{A}$) = **true**]                      ▷ Takes long
 6:     **Return** $\mathcal{A}$

---

In order to do so, it first constructs an arbitrary matching $\Pi$ of the points. A matching $\Pi$ is a set of pairs $(x, y)$ where $x, y \in X$, it also partitions $X$ into a set of $\frac{|X|}{2}$ disjoint pairs. Given this matching, this algorithm randomly picks one of the points in each pair, removing the other point of the pair, resulting in a subset of remaining points $X_1 \subset X$ where $|X_1| = \frac{|X|}{2}$.

Continuing this process $k$ times for the following value of $k$

$$2^k = \frac{|X|}{c_0 \frac{\delta}{\varepsilon} \log \frac{\delta}{\varepsilon}} \tag{2}$$

results in the set $|X_k|$ which is an $\varepsilon$-net for $X$. It is easy to make this process deterministic by following the conditional expectation method at each halving step [6].

---

**Algorithm 3** Building $\varepsilon$-net (Discrepancy-based Algorithm)

---

**Require:** The range space $(X, \mathcal{R})$, value of $\varepsilon$, failure probability $\varphi'$.
**Ensure:** The $\varepsilon$-net $\mathcal{A}$.
 1: **function** BUILDEPSNETDISC($X, \varepsilon, \mathcal{R}$)              ▷ Build $\varepsilon$-net by providing $\varepsilon$
 2:     $k \leftarrow$ number of iterations (Equation 2)
 3:     $\mathcal{A} \leftarrow X$
 4:     **for** $1 \leq i \leq k$ **do**
 5:        $\mathcal{A} \leftarrow Halving(\mathcal{A}, \mathcal{R})$          ▷ The halving step, with arbitrary matching.
 6:     **Return** $\mathcal{A}$

---

**Comparison.** The randomized algorithm is straightforward to implement, requiring only random sampling from each layer $\mathcal{L}_i$ to construct the subsequent layer $\mathcal{L}_{i+1}$. However, it is inherently randomized and provides running-time guarantees in expectation. Furthermore, its time complexity depends on the time to verify if the selected set is indeed an $\varepsilon$-net.

In contrast, the deterministic discrepancy-based algorithm deterministically constructs an $\varepsilon$-net by progressively halving each layer $\mathcal{L}_i$. This process involves only halving $\mathcal{L}_i$ a couple of times to identify the next layer $\mathcal{L}_{i+1}$ (See BuildEpsNetDisc in Algorithm 3). Nevertheless, the running time of the discrepancy-based algorithm depends exponentially on the dimensionality of the input points, which makes it impractical.

## B.2 BUDGET-AWARE ALGORITHM FOR CONSTRUCTING $\varepsilon$-NETS

To bridge the gap between sampling-based methods (fast but probabilistic) and discrepancy-based methods (deterministic but computationally expensive), we introduce a *budget-aware algorithm*. This approach allows the user to control a resource budget, thereby adjusting the trade-off between construction speed and the probability that the resulting set forms a valid $\varepsilon$-net.

Given a user-specified timing budget $\mathcal{B}$, which determines the allowed construction time, we design an algorithm that achieves success with a probability depending on $\mathcal{B}$. This approach is inspired by the deterministic $\varepsilon$-net construction via finding an unhit range, known as NET-FINDER algorithm [44]. Our proposed BUDGET-AWARE algorithm is presented in Algorithm 4.

---

**Algorithm 4** Budget-Aware $\varepsilon$-net Construction

---

**Require:** The range space $(X, \mathcal{R})$, value of $\varepsilon$, the budget $\mathcal{B}$.
**Ensure:** The $\varepsilon$-net.
1: **function** BUDGETAWARE$(X, \varepsilon, \mathcal{B})$
2:     $\mathcal{N} \leftarrow$ small random sample from $X$.                           ▷ See [44]
3:     **for** $i \leq \mathcal{B}$ **do**                                    ▷ At most $\mathcal{B}$ iterations.
4:         $R \leftarrow$ FindUnhitRange$(\mathcal{N}, \mathcal{R})$      ▷ Find $R \in \mathcal{R}$ where $|R| \geq \varepsilon \cdot |X|$ but $R \cap \mathcal{N} = \varnothing$.
5:         **if** $R$ exists **then**
6:             Add $O(1)$ random points from $R$ to $\mathcal{N}$.
7:         **else**
8:             break                             ▷ $\mathcal{N}$ is an $\varepsilon$-net.
        **return** $\mathcal{N}$

---

The algorithm begins with a small random sample $\mathcal{N}$ from the point set $X$, similar to GENERAL NET-FINDER [44]. At each iteration, it identifies a *heavy* range $R$ not intersecting $\mathcal{N}$:

$$R \in \mathcal{R}, \quad |R| \geq \varepsilon \cdot |X|, \quad R \cap \mathcal{N} = \varnothing.$$

While GENERAL NET-FINDER proceeds until all heavy ranges are covered, BUDGET-AWARE terminates after at most $\mathcal{B}$ iterations. The parameter $\mathcal{B}$, specified by the user, controls the preprocessing time and introduces a trade-off between efficiency and the final success probability (see below).

### B.3 ANALYSIS OF BUDGET-AWARE ALGORITHM

We analyze both the running time of the BUDGET-AWARE algorithm and bound the failure probability as a function of the user-provided budget $\mathcal{B}$.

**Success Probability.** Let $\mathcal{C}$ denote the random variable representing the number of calls to the oracle FindUnhitRange (Line 3) needed to obtain an $\varepsilon$-net. Prior work [44] shows that $\mathbb{E}[\mathcal{C}] = O(1/\varepsilon)$. Failure occurs when $\mathcal{C} \geq \mathcal{B}$, meaning that $\mathcal{B}$ calls are not enough, hence by Markov's inequality:

$$\Pr(\texttt{failure}) = \Pr(\mathcal{C} \geq \mathcal{B}) \leq \frac{\mathbb{E}[\mathcal{C}]}{\mathcal{B}} \leq \frac{1}{\varepsilon \cdot \mathcal{B}}.$$

Thus, increasing $\mathcal{B}$ improves success probability: doubling $\mathcal{B}$ halves the failure probability.

**Time Complexity.** Each iteration (Line 3 in Alg. 4) requires one call to FindUnhitRange. A naive implementation checks all $R \in \mathcal{R}$ for intersection with $\mathcal{N}$, costing $O(|\mathcal{R}|) = O(n^\delta)$ for VC-dimension $\delta$. To improve practicality, in our experiments, we instead partition the space into disjoint ranges of size at least $\varepsilon n$, searching only within this partition. This heuristic reduces the cost to $O(n)$ per call.[11] Therefore, the total preprocessing time is $O(\mathcal{B} \cdot n)$.[12] See experiments for more details (Section 7 and Appendix I).

**Output Size.** The output of NET-FINDER is known to be an $\varepsilon$-net of size $O(\frac{d}{\varepsilon} \log \frac{1}{\varepsilon})$ (see Theorem 1). The same bound holds for the BUDGET-AWARE algorithm.

## C BACKGROUND ON PROXIMITY GRAPHS

Graph-based algorithms for the ANN problem typically begin by constructing a graph on the given dataset $X$. A key property of these graphs is *navigability*, which ensures that the Greedy Search algorithm (Algorithm 5) can be effectively applied [41]. Specifically, navigability means that by following a sequence of locally greedy steps, the algorithm can successfully reach an approximate nearest neighbor of the query point $q$.

---

[11]With optimized libraries and parallelization, this step can be made faster in practice.

[12]Our success probability analysis assumes the naive implementation, which always identifies an unhit range. In practice, however, we show experimentally that the heuristic variant also achieves reliable performance.

According to this definition, a **complete graph** over the point set is trivially navigable. The most optimized navigable graph can, in principle, be obtained by constructing the dual of the Voronoi diagram of the points, known as the **Delaunay triangulation** [9]. However, constructing this graph is computationally challenging, particularly in high dimensions, due to the curse of dimensionality.

An efficient approximation of the Delaunay triangulation can be achieved through a simple randomized algorithm that incrementally inserts points and connects each new point to its nearest neighbors in the existing graph structure. This approach forms the basis of the **Navigable Small World** (NSW) graph [41]. The HNSW algorithm adopts a similar strategy within each layer, while introducing additional heuristics to improve practical performance, such as adding random exploration edges between points. These heuristics can also be incorporated into the HENN structure as well, treating the navigable graph as a black-box [40].

A natural baseline is the $k$-**NN graph**, where each point is connected to its $k$ nearest neighbors. While this structure is easy to build and widely used, it is well known that for small values of $k$, $k$-NN graphs tend to exhibit numerous local minima that hinder greedy navigation [37].

Another classical proximity structure is the **Relative Neighborhood Graph (RNG)**, introduced in computational geometry [51]. An edge between two points exists only if no other point lies within the lens defined by them, making the RNG a sparse subgraph of the Delaunay triangulation. While RNGs enjoy theoretical navigability guarantees, their practical construction cost and sparsity often limit their use in large-scale ANN systems.

The **Navigable Spreading-Out Graph (NSG)** [14] has emerged as a practical ANN graph structure that carefully sparsifies a $k$-NN graph while ensuring connectivity and navigability. NSG uses a diversification step to spread out edges and eliminate redundancy, resulting in a graph that balances efficiency and search quality. It is among the most competitive structures in large-scale ANN benchmarks.

For a more detailed comparison of these graph structures, we refer the reader to [57]. As highlighted earlier, any navigable graph can be treated as a black box and seamlessly integrated into the HENN framework within each layer.

## D  DISCUSSION AND LIMITATIONS

We highlight several practical aspects and limitations of constructing general HENN graphs using Algorithm 1.

HENN construction can be naturally *parallelized* during the indexing phase, enabling faster preprocessing. Moreover, it can support *dynamic updates* to the dataset, provided that the layers are maintained and the $\varepsilon$-net property is preserved. Further details on parallel construction and dynamic maintenance are given in Appendices F and E.

HENN is defined as a structural property that extends to any metric $\mathbf{d}$ yielding bounded VC-dimension for ring ranges. This includes widely used metrics such as $\ell_p$-norms and angular distances (cosine similarity). However, if the VC-dimension is unbounded, $\varepsilon$-net construction becomes impractical.

The theoretical time guarantees of HENN rely on properties of the underlying proximity graphs, particularly their degree and recall bounds. In practice, many commonly used proximity graphs exhibit constant degree and bounded recall. However, certain graphs, such as the Delaunay Triangulation (DT), may have degrees that grow exponentially with dimension.

## E  DYNAMIC SETTING

In this section, we present a procedure for maintaining the HENN structure under dynamic updates to the point set $X$. The supported operations include `Insert(x)`, which adds a new point $x$, and `Delete(x)`, which removes an existing point $x \in X$.

Based on the discussion in Theorem 1, a random sample of an appropriate size forms an $\varepsilon$-net of $X$ with high probability. We denote this required sample size by $m_\varepsilon$, given by:

$$m_\varepsilon = O\left(\frac{d}{\varepsilon} \log \frac{d}{\varepsilon}\right)$$

Each layer of HENN, denoted by $\mathcal{L}_i$, is constructed as a random sample of size $m_{\mathcal{E}(|\mathcal{L}_{i-1}|)}$ from the previous layer (see Equation 1). Consequently, the problem reduces to dynamically maintaining a random sample $S$ of size $m_\varepsilon$ from the point set $X$.[13]

This problem can be addressed using Reservoir Sampling [53] and the *Backing Samples* technique [17]. The key idea is to handle `Insert(x)` operations by probabilistically adding the new element to the sample $S$ using a non-uniform coin toss. To support deletions, a larger backing sample is maintained beyond size $m_\varepsilon$, allowing for efficient resampling of $S$ once the size drops below a threshold. This approach yields a constant amortized update time.

According to this, we can maintain the HENN structure dynamically:

**Insert(x):** To insert a new point, dynamic updates are performed starting from layer $\mathcal{L}_1$ and proceeding upward through the hierarchy, stopping at the highest layer where the new point is included. This process takes $O(\log n)$ time, matching the insertion time complexity of HNSW.

**Delete(x):** Deletion begins at layer $\mathcal{L}_1$, where the point $x$ is removed if present. If the size of a layer falls below a critical threshold (as discussed in [17]), the layer must be resampled. Following resampling, the HENN structure is rebuilt from that layer up to the root, which incurs a cost of $O(n \log n)$ in the worst case. However, since such rebuilding occurs infrequently, only when the layer size drops significantly (e.g., $m_\varepsilon < c_0 n$ for a constant $c_0$), the amortized cost remains $O(\log n)$.

## F    PARALLELIZATION

In this section, we present a parallelized approach to constructing the HENN index during pre-processing. While the original HENN construction, shown in Algorithm 1, runs sequentially by building layers $\mathcal{L}_1$ through $\mathcal{L}_m$ from the base point set $X$, this process can be parallelized to reduce preprocessing time.

To enable parallelization, we exploit the fact that layer sampling in HENN is performed with replacement. Given $p$ parallel CPU cores, we can independently generate samples for each layer in parallel. Specifically, each core performs independent sampling, effectively achieving a $p$-factor speedup for the sampling phase performed on each layer.

HNSW also uses a parallelization to enhance preprocessing [40], where each input point is processed independently. For each point, a random level is assigned, and the point is inserted into the corresponding layers. This results in a total construction time of $O(n \log n)$, which can be reduced to $O(\frac{n \log n}{p})$ under parallel execution with $p$ cores.

For HENN, the construction begins by sampling a subset of size $\frac{n}{2^m}$ for the first layer $\mathcal{L}_1$, and recursively building higher layers. Each layer $\mathcal{L}_i$ requires constructing a navigation graph, the complexity of which depends on the chosen method. For instance, NSW-based graph construction requires $O(|\mathcal{L}_i| \log |\mathcal{L}_i|)$ time on the $i$th layer. Excluding graph construction, the sampling phase alone can be executed in $O(\frac{n \log n}{p})$ time using $p$ cores.

## G    PROOFS

### G.1    PROOF OF LEMMA 3

**Proof**    *Since $\mathcal{L}_1$ is an $\varepsilon$-net of $\mathcal{L}_0$, we know that each (ring) range $R$ of size more than $\varepsilon \cdot n$, intersects with $\mathcal{L}_1$. In other words:*

$$|R| \geq \varepsilon n \to \mathcal{L}_1 \cap R \neq \varnothing$$

---

[13] $S$ is a random sample with replacement, with each element having a probability $\frac{m_\varepsilon}{n}$ being in $S$.

*This comes from the definition of an $\varepsilon$-net.*

*Based on the definition of recall bound, we know that there are at most $\rho_\gamma$ more points, denoted as $P = \{p_1, p_2, \cdots, p_{\rho_\gamma}\}$, in $\mathcal{L}_1$ that are closer to $q$ than $p$ (with probability of at least $\gamma$).*

*Assume that the points in $P$ are sorted based on their distance to $q$, with $p_1$ being the closest. For each $p_j \in P$ define a unique range $R_j$, which is a ring centered at $q$, covering all the distances between $\big(\mathbf{d}(q, p_{j-1}), \mathbf{d}(q, p_j)\big)$, exclusively (see Figure 8). In addition, define one more ring, $R_{\rho_\gamma + 1}$ for $\big(dist(q, p_{\rho_\gamma}), dist(q, p)\big)$.*

*All these $\rho_\gamma + 1$ ranges are disjoint, and they do not contain any point in $\mathcal{L}_1$. Since $\mathcal{L}_1$ is an $\varepsilon$-net for this range space, this means that all the ranges $R_j$ have at most $\varepsilon \cdot n$ points from $\mathcal{L}_0$ (the lower level). As a result, the union of all these ranges contains at most $\varepsilon \cdot n \cdot (\rho_\gamma + 1)$ points.* □

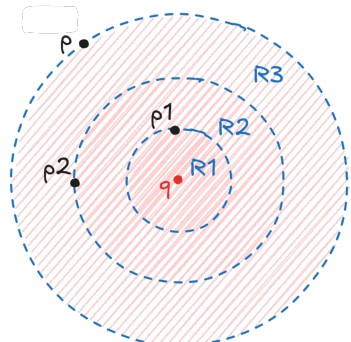

Figure 8: Visualization of Lemma 3. In this example, $\rho_\gamma = 2$ and the black points are inside the $\varepsilon$-net.

### G.2   PROOF OF THEOREM 4

**Proof**   *The query process begins at the top layer $\mathcal{L}_L$, where GS is executed on the corresponding proximity graph. The search then proceeds layer by layer until reaching the base layer $\mathcal{L}_0$.*

*In the final step, the algorithm identifies a point*

$$p = \mathsf{GS}(\mathcal{G}_{\mathcal{L}_1}, s),$$

*the output of greedy search on $\mathcal{G}_{\mathcal{L}_1}$ starting from some initial node $s$. From $p$, the search continues in the base layer $\mathcal{L}_0$. We analyze the running time inductively, proceeding from the base layer upward.*

*Let $T(n)$ denote the time required to query a HENN graph constructed on a dataset of size $n$. Computing $p$ requires $T(|\mathcal{L}_1|)$ time, since layers $\mathcal{L}_1, \ldots, \mathcal{L}_L$ themselves form a HENN structure. By Lemma 3, continuing the search from $p$ in the base layer requires at most $O(\rho_\gamma \cdot \varepsilon n)$ hops. If $\pi$ is the degree of the proximity graph, the total cost per hop is $O(\pi)$. Thus,*

$$T(n) = T(|\mathcal{L}_1|) + O(\pi \rho_\gamma \varepsilon n).$$

*Treating $\rho_\gamma$ and $\pi$ as fixed constants gives*

$$T(n) = T(m_{\mathcal{E}(n)}) + O(\varepsilon n),$$

*where $m_\varepsilon$ is the size of the $\varepsilon$-net. By Theorem 1 and using the fact that VC-dim is $\Theta(d)$, we have $m_\varepsilon = O\big(\frac{d}{\varepsilon} \log \frac{1}{\varepsilon}\big)$. Substituting this bound yields*

$$T(n) = T\big(O\big(\tfrac{d}{\varepsilon} \log \tfrac{1}{\varepsilon}\big)\big) + O(\varepsilon n).$$

*Ignoring constant factors, we obtain*

$$T(n) \le T\big(\tfrac{d}{\varepsilon} \log \tfrac{1}{\varepsilon}\big) + \varepsilon n.$$

*Let $x(n) = \frac{1}{\varepsilon}$ as a function of $n$. Then,*

$$T(n) \le T\big(d \cdot x(n) \log x(n)\big) + \frac{n}{x(n)}.$$

*To guarantee convergence, we require*

$$d \cdot x(n) \log x(n) = o(n).$$

*At the same time, to minimize the second term $\frac{n}{x(n)}$, we select $x(n)$ as large as possible subject to this constraint. The optimal choice is obtained by solving*

$$x(n) \log x(n) = \tfrac{n}{d},$$

*which admits the asymptotic solution*

$$\log x(n) = W(\tfrac{n}{d}) \approx \log(n/d) - \log \log(n/d) + O(1),$$

*where $W$ denotes the Lambert $W$ function [33]. Consequently, up to lower-order terms,*

$$x(n) = \tfrac{n}{d \log n}.$$

*This implies*

$$\tfrac{n}{x(n)} = d \cdot \log n,$$

*and therefore, since $L = \log n$, the total running time is (bringing back the constants $\rho_\gamma$ and $\pi$)*

$$T(n) = O(\rho_\gamma \pi d \log^2 n).$$

*Finally, to see why this choice of $x(n)$ is optimal, suppose instead we take $x(n)$ such that*

$$d \cdot x(n) \log x(n) = n^{1-\alpha}, \quad \alpha > 0.$$

*Then $x(n) \approx \frac{n^{1-\alpha}}{d \log n}$, which yields*

$$\tfrac{n}{x(n)} = dn^\alpha \log n,$$

*leading to a runtime strictly larger than $O(d \log^2 n)$. Hence, the choice $x(n) = \frac{n}{d \log n}$ is asymptotically optimal.*

*To achieve this running time, two conditions must hold: (1) each layer must form an $\varepsilon$-net of the preceding one, and (2) the greedy search at each layer must return a point within the $\rho_\gamma$ nearest neighbors of the query in the layer below.*

*For (1), since each layer is an $\varepsilon$-net with probability at least $\varphi$, the probability that all $L$ layers are valid $\varepsilon$-nets is $\varphi^L$. For (2), by the definition of the recall bound, the event occurs with probability at least $\gamma$ per layer, and thus $\gamma^L$ across all layers. Combining these independent events, the above query time is achieved with probability at least $(\varphi \cdot \gamma)^L$ for $L = \log n$.* □

## G.3 PROOF OF LEMMA 5

**Proof** *For a large enough constant $c_1 \geq c$, choosing $\varepsilon = \frac{c_1 \delta \log n}{n}$ we have:*

$$\frac{8\delta}{\varepsilon} \log \frac{16}{\varepsilon} \leq \frac{n}{c}.$$

*Choose a failure probability $\varphi' = \frac{\log n}{c_2 n}$ for a large enough constant $c_2 > 1$. Then,*

$$\frac{4}{\varepsilon} \log \frac{4}{\varphi'} \leq \frac{8\delta}{\varepsilon} \log \frac{16}{\varepsilon}.$$

*Hence, by Theorem 1, we obtain*

$$\max\left\{ \tfrac{4}{\varepsilon} \log \tfrac{4}{\varphi'}, \ \tfrac{8\delta}{\varepsilon} \log \tfrac{16}{\varepsilon} \right\} = \tfrac{8\delta}{\varepsilon} \log \tfrac{16}{\varepsilon} \ \leq \ \tfrac{n}{c}.$$

*Therefore, setting $m_\varepsilon = \frac{n}{c}$ gives us that $\mathcal{L}_{i+1}$ is an $\varepsilon$-net of $\mathcal{L}_i$, with*

$$\varepsilon = \Theta\left(\tfrac{\delta \log n}{n}\right) \quad \text{and success probability at least} \quad 1 - \Theta\left(\tfrac{\log n}{n}\right).$$

□

---

**Algorithm 5** Greedy Search Algorithm

---

**Require:** The HENN graph $\mathcal{H}$ and the query point $q$.
**Ensure:** The approximate nearest neighbor of $q$ in $X$.
1: **function** QUERY($\mathcal{H}, q$)
2:      $v \leftarrow$ *random point from* root($\mathcal{L}_0$)
3:      **for** each layer $i = L, L-1, \cdots, 0$ **do**
4:          $v \leftarrow$ *GreedySearch*($\mathcal{L}_i, v, q$)
5:      **Return** $v$
6: **function** GREEDYSEARCH($\mathcal{G}, v, q$)        $\triangleright$ $\mathcal{G}$ is the proximity graph, starting node $v$, and query $q$
7:      $next \leftarrow v$
8:      **repeat**
9:          $curr \leftarrow next$
10:         $\mathcal{N} \leftarrow$ neighbors of $curr$ in $\mathcal{G}$.
11:         $next \leftarrow \arg\min_{u \in \mathcal{N}} \mathbf{d}(u, q)$                $\triangleright$ Closest neighbor to $q$
12:      **until** $\mathbf{d}(next, q) \geq \mathbf{d}(curr, q)$        $\triangleright$ Until getting stuck in local minimum
13:      **Return** $curr$

---

## H   PSEUDO-CODES

## I   MORE ON EXPERIMENTS

### I.1   DETAILS ON EXPERIMENTAL SETTING

**Real Datasets.** We evaluate our methods on standard ANN benchmarks with both Euclidean and angular (cosine) distance metrics. The datasets include: SIFT-128 [28], consisting of 1M vectors in 128 dimensions; GIST-960 [28], with 1M vectors in 960 dimensions; FASHION-MNIST, containing 60K vectors in 784 dimensions; and NYTIMES, with 290K vectors in 56 dimensions. Among these, NYTIMES is evaluated with cosine similarity, while the others use the Euclidean norm.

**Synthetic Datasets.** To study the effect of varying parameters such as dimensionality, we generated synthetic datasets for ANN search. These were drawn either from a uniform distribution or from mixtures of anisotropic Gaussians, producing several skewed clusters that mimic the structure of challenging real-world datasets.

**Queries.** For our experiments, queries are generated in two ways: by randomly sampling a subset of points from the dataset, and by creating an additional set of points drawn uniformly at random from the ambient space.

**Methods.** In our experiments, we combined HENN with different PGs at each level. For this purpose, we followed the construction algorithm 1 with a standard construction process for the PG.

**Implementation of $\varepsilon$-net Construction.** We employ two approaches for building $\varepsilon$-nets. The first is *Random Sampling*, where a random subset of size $m_\varepsilon$ is selected, which yields an $\varepsilon$-net with a certain probability. Other approaches, such as sketch-and-merge or discrepancy-based methods, are not practical in this context as their complexity grows exponentially with the dimension $d$. Our second approach is the *Budget-Aware Algorithm*, in which the user specifies a budget $\mathcal{B}$, expressed as a ratio $r \in (0, 1]$.

Concretely, if the target $\varepsilon$-net size is $m$, we first select $(1-r)m$ elements uniformly at random. The remaining $r \cdot m$ elements are then chosen using a heuristic procedure (see Algorithm 6), which is based on the unhit-set discovery technique described in Section B.1.

### I.2   COMPARISON OF PROXIMITY GRAPHS

In this section, we compare several proximity graphs, including KNN, KGRAPH, NSG, FAANG, and NSW. While this is not the primary focus of our work, we report their recall bounds and query times in a flat (single-layer) setting across multiple datasets.

---

**Algorithm 6** Finding an Unhit Range

---

**Require:** A point set $X$ and a subset $\mathcal{N}$ as the $\varepsilon$-net placeholder.
**Ensure:** The $\varepsilon$-net with high probability.
 1: **function** FINDANUNHITSET($X, \varepsilon, \mathcal{N}$)
 2:     **for** $i \leq \mathcal{B}$ **do**                                          ▷ At most $\mathcal{B}$ iterations.
 3:         $p \leftarrow$ a random point from $X$
 4:         Partition the whole space based on the distance of points to $p$.
 5:         **for** partition $P_i$ **do**
 6:             **if** $P_i \cap \mathcal{N} = \varnothing$ **then**
 7:                 Add a random point from $P_i$ to $\mathcal{N}$
 8:             If no new points added, just add a random point to $\mathcal{N}$.
        **return** $\mathcal{N}$

---

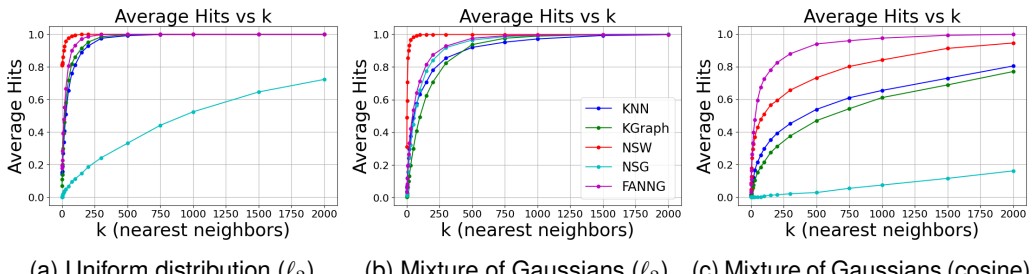

(a) Uniform distribution ($\ell_2$)    (b) Mixture of Gaussians ($\ell_2$)    (c) Mixture of Gaussians (cosine)

Figure 9: Comparison of recall bound (average hit among $k$ neighbors of the query) between different proximity graphs (synthetic dataset).

Figures 9 and 10 present the recall bounds of these proximity graphs on both synthetic and real datasets. In these experiments, we varied the value of $k$ in the $k$-nearest neighbor search and, over multiple runs, measured the fraction of queries for which the returned neighbors included the true $k$ nearest neighbors. This fraction provides an estimate of the probability of correctly retrieving the $k$ nearest neighbors. As $k$ increases, this probability naturally improves, since the search is more likely to include points from the true neighborhood of the query. We define the recall bound of a proximity graph (Definition 3) as the smallest value of $k$ for which this probability exceeds 0.9 on average. Notably, some graphs, such as NSW (used in HNSW), show very small recall bounds, which in turn leads to better performance for HENN.

Figures 12 and 11 compare the query-time performance of these proximity graphs in a single-layer (flat) setting across different datasets.

## I.3    INDEXING SIZE AND TIME

In this section, we evaluate the indexing phase of HENN, focusing on index size and preprocessing time. Figure 13 reports the indexing cost of HENN when integrated with different proximity graphs. We observe that the build time for NSW is higher than for other graphs, since our implementation includes a second refinement phase to enforce the desired node degree specified by the user. In terms of memory usage, the index grows almost linearly with both the dataset size $n$ and the dimensionality $d$.

## I.4    ADDITIONAL EXPERIMENTS

**Effect of $n$ and $d$.**    Figure 14 illustrates the effect of dataset size $n$ on query time, while Figure 15 reports its effect on the number of hops (steps) in the greedy search. In both cases, the dependence on $n$ is logarithmic, consistent with Theorem 4. Figure 16 shows the effect of dimensionality $d$ on query time and hops, which scales linearly with $d$ as predicted by Theorem 4. As discussed earlier, increasing $d$ raises the likelihood of getting trapped in local minima (for proximity graphs with fixed degree). Consequently, for smaller values of $d$, the query time initially decreases before the linear

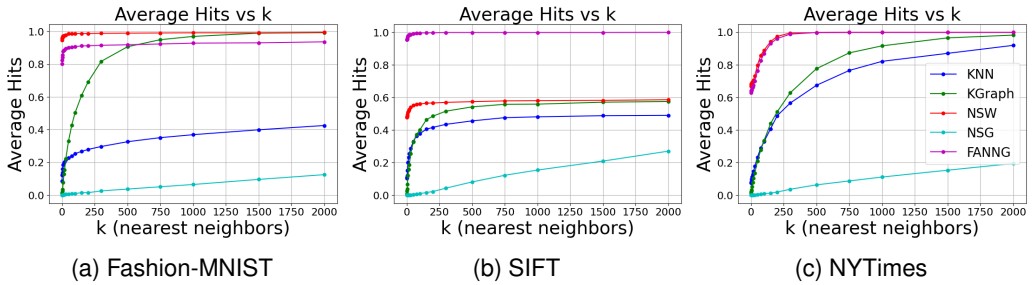

(a) Fashion-MNIST          (b) SIFT          (c) NYTimes

Figure 10: omparison of recall bound (average hit among $k$ neighbors of the query) between different proximity graphs (real datasets).

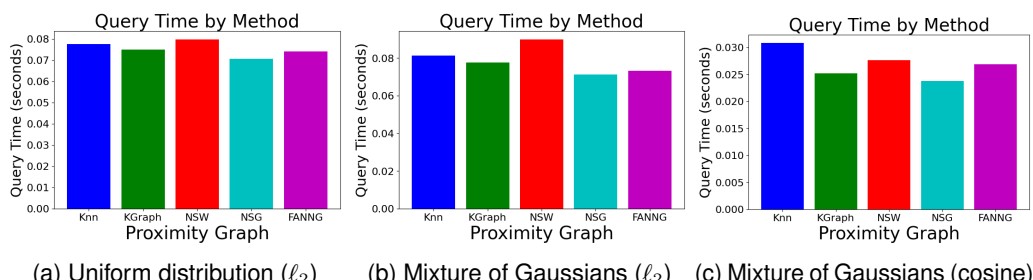

(a) Uniform distribution ($\ell_2$)    (b) Mixture of Gaussians ($\ell_2$)    (c) Mixture of Gaussians (cosine)

Figure 11: Comparison of query times of different PGs in a single-layer setting (synthetic dataset).

dependence on $d$ dominates. Figure 17 also shows the effect of dimension $d$ on the synthetic data with Mixture of Gaussians distribution.

**Budget-Aware $\varepsilon$-net construction**    Figure 18 demonstrates the effectiveness of the budget-aware algorithm in constructing $\varepsilon$-nets with high probability. Across multiple real datasets, even a small preprocessing budget (as low as 10%) substantially increases the likelihood that the resulting set forms a valid $\varepsilon$-net. This property is particularly important for **compressing the HENN index**: by selecting smaller subsets per layer, we can reduce the overall index size, but at the cost of recall, since the subsets may no longer be $\varepsilon$-nets with high probability. Incorporating the budget-aware strategy mitigates this issue, yielding smaller subsets that retain a high probability of being $\varepsilon$-nets and thus achieve recall comparable to the non-compressed version (as shown earlier in Figure 6).

Figure 19 further illustrates the trade-off in preprocessing time. While the budget-aware algorithm requires additional time, growing linearly with the budget, it provides substantially better $\varepsilon$-net quality, making the extra preprocessing cost as a trade-off.

**Effect of Proximity Graph Degree.**    Figure 20 illustrates the impact of varying the degree of the proximity graph (across all layers) on the query time of HENN over additional datasets. The results confirm a linear dependence on the graph degree.

**Effect of exponential decay**    Figure 21 shows the effect of varying the exponential decay rate, which controls the reduction in layer size (and equivalently, the number of layers) on recall and query time. Smaller decay values produce deeper hierarchies with more layers, leading to higher recall but longer query times, since the greedy search must traverse additional layers. Interestingly, there exists an optimal number of layers that minimizes query time. Increasing the decay (reducing the number of layers) weakens the hierarchical structure and degrades performance. In the extreme case of very few layers, query time increases again, as most of the effort is spent in the top layer, which contains $O(n)$ points, making it more prone to getting stuck in local minima.

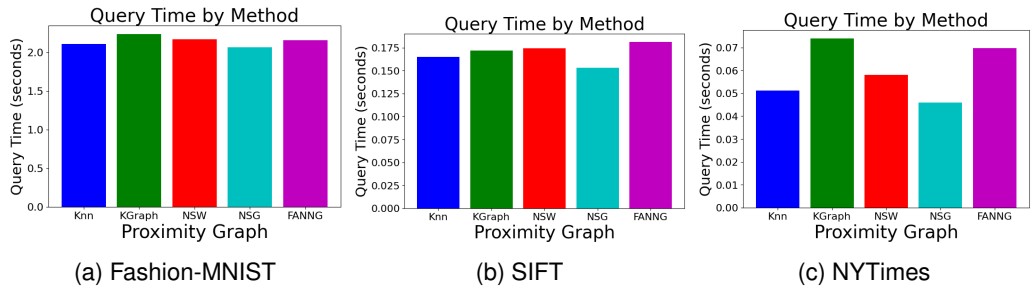

(a) Fashion-MNIST  (b) SIFT  (c) NYTimes

Figure 12: Comparison of query times of different PGs in a single-layer setting (real dataset).

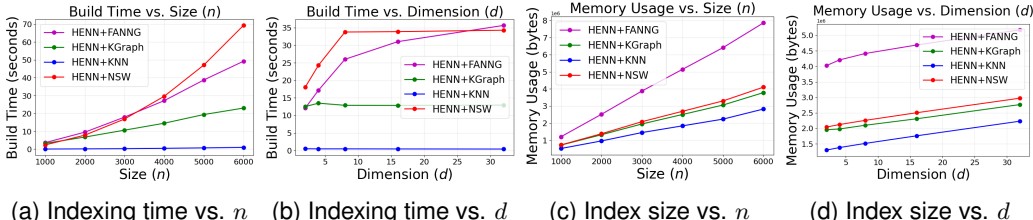

(a) Indexing time vs. $n$  (b) Indexing time vs. $d$  (c) Index size vs. $n$  (d) Index size vs. $d$

Figure 13: Index size and index time as a function of dataset size $n$ and dimension $d$. This is the synthetic dataset with Mixture of Gaussians distribution.

## J   USE OF LARGE LANGUAGE MODELS (LLMS)

LLMs were used for polishing text, checking grammar, and assisting in debugging code. All research ideas, methods, experiments, and analyses are the authors' own. The authors take full responsibility for the content.

## K   MORE ON RECALL BOUND

In this section, we further examine the recall bound parameter introduced in Definition 3 and provide deeper insight into the results of Theorem 4. In particular, we interpret the theorem's implications: broadly, Theorem 4 derives a probability distribution for the final running time (i.e., the number of visited hops) in the HENN index. This distribution is governed by the structural properties of the underlying proximity graph.

Let $K$ be a *random variable* denoting the rank of the output returned when running the greedy search algorithm on the chosen proximity graph $\mathcal{G}$. In other words, a search over $\mathcal{G}$ returns the $K$-th nearest neighbor of the query point. The distribution of $K$ is inherently determined by the structure and quality of the proximity graph: a better-designed graph places more probability mass on smaller values of $K$, yielding a lower expected rank.

The exact distribution of this variable can be empirically observed by repeatedly sampling from it, that is, by running the search algorithm and recording the quality of its outputs (see Figures 9 and 10). While our goal in this paper is not to derive closed-form distributions for different proximity graphs, doing so would be an interesting direction for future work.

We define the PMF and CDF of $K$ as

$$p_k := \Pr(K = k), \qquad F_K(k) := \Pr(K \leq k) = \sum_{j=1}^{k} p_j.$$

Based on our definition of the recall bound (Definition 3), the value $\rho_\gamma$ is given by:

$$\rho_\gamma := F_K^{-1}(\gamma).$$

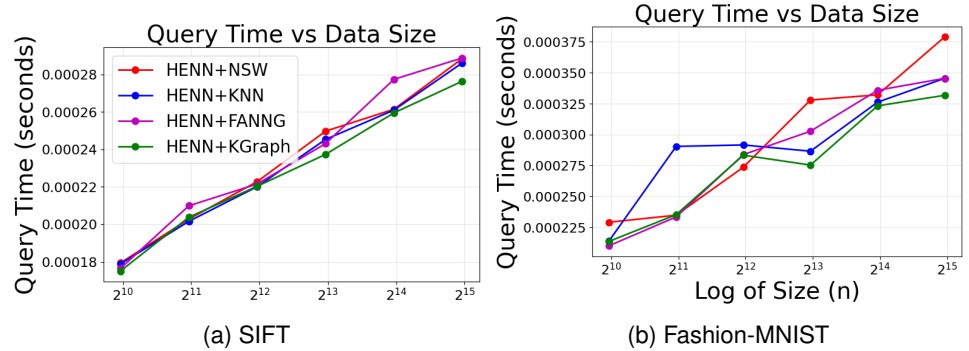

Figure 14: Effect of dataset size $n$ on the query time.

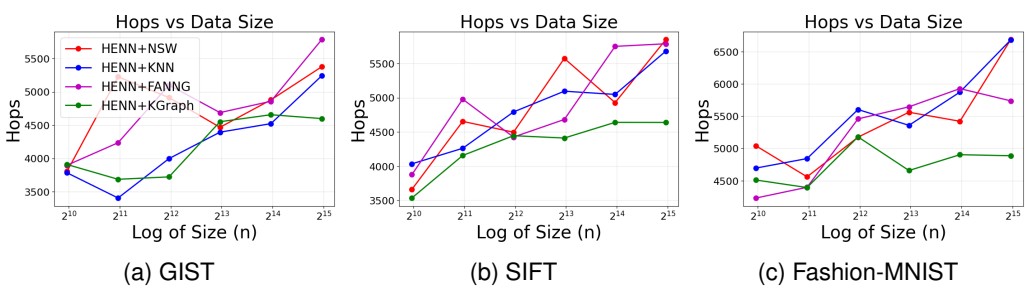

Figure 15: Number of visited hops vs $n$.

Following Theorem 4, for any fixed value $K = k$, the running time of HENN is

$$O(k \cdot d \cdot \log^2 n)$$

with probability at least $p_k^L$, where $L$ is the number of layers, which is at most $O(\log n)$.

Let $T$ be the random variable representing the number of hops visited (i.e., the running time of HENN). Theorem 4 therefore implies the following: for any $k \in \mathbb{N}$ and a fixed constant $c > 0$, defining

$$t_k := ckd\log^2 n \qquad \text{and} \qquad L := \log n,$$

the running-time random variable $T$ satisfies

$$\Pr(T \le t_k) = \Pr(T \le ckd\log^2 n) = \big(F_K(k)\big)^L.$$

Thus, the CDF of $T$ on the discrete grid $\{t_k\}_{k\geq 1}$ is

$$F_T(t_k) := \Pr(T \le t_k) = \big(F_K(k)\big)^L, \qquad k = 1, 2, \ldots,$$

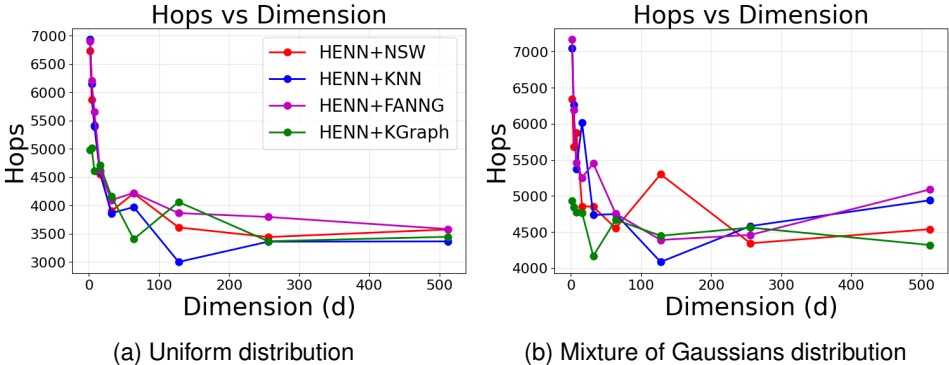

Figure 16: Number of visited hops vs. $d$ (synthetic dataset)

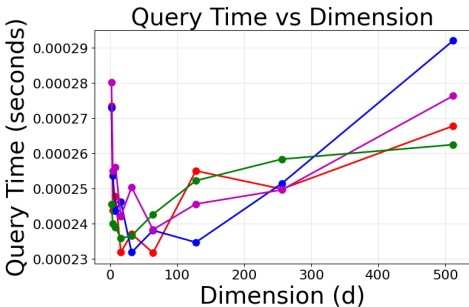

Figure 17: Effect of dimension $d$ on the query time. The results are on the synthetic dataset with varying $d$ (Mixture of Gaussians distribution).

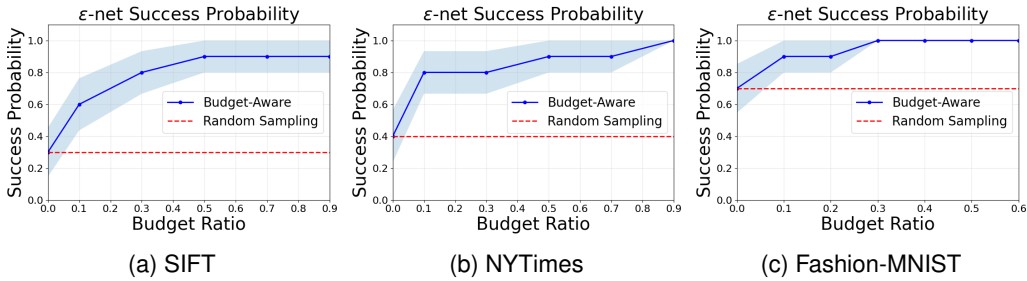

(a) SIFT  (b) NYTimes  (c) Fashion-MNIST

Figure 18: The comparison of Random Sampling and Budget-Aware algorithms on finding an $\varepsilon$-net. The budget ratio $r$ means that for an $\varepsilon$-net of size $m_\varepsilon$, we added $(1 - r) \cdot m_\varepsilon$ points greedily by finding unhit ranges (see Appendix I for details). The experiment is run on small subset of these datasets.

with $F_T(t_0) := 0$.

From this, we can derive the probability mass function of the running-time distribution of HENN:

$$\Pr(T = t_k) = \big(F_K(k)\big)^L - \big(F_K(k-1)\big)^L. \tag{3}$$

**Result.**  Based on the above, Theorem 4 provides a probability distribution for the running time of the HENN structure, expressed in terms of the properties of the underlying proximity graph. Since each proximity graph induces its own distribution over the random variable $K$, graphs whose distributions place more mass on smaller values of $K$ (e.g., those with smaller medians) correspondingly yield better average running times.

### K.1  EMPIRICAL RESULTS

In this subsection, we present empirical results to validate the preceding discussion. We begin by examining the distribution of $K$ across different proximity graphs. Then, using equation 3 (derived from Theorem 4), we compute the corresponding distribution of the HENN running time. Finally, we compare these theoretical predictions with empirical measurements to confirm that the observed running times of HENN align with these insights. In our experiments, we report the expected running time and compare it across different proximity graph constructions.

**Settings.**  We conducted experiments using the following proximity graphs: KNN, KGraph, NSW, and FAANG. The datasets used were `Random (L2)`, `Mixture of Gaussians L2`, `NYTimes (Cosine)`, and `GloVe (Cosine)`. Additional details about these datasets and experimental configurations are provided in the main Experiments section of the paper.

**Distribution of $K$.**  Figure 22 illustrates the empirical distribution of $K$ for the different proximity graphs. For example, on the `Mixture of Gaussians` dataset, several graphs tend to get trapped in local minima, which creates a secondary peak (a local mode) in the middle of the distribution.

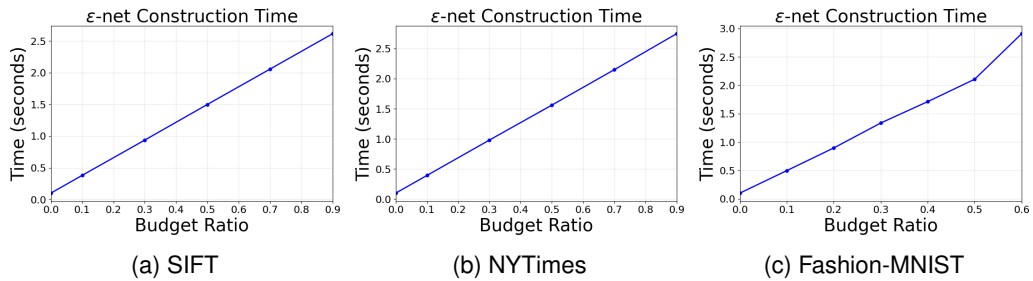

Figure 19: Comparing the time spent to find $\varepsilon$-net using Budget-Aware algorithm for different values of budget $\mathcal{B}$.

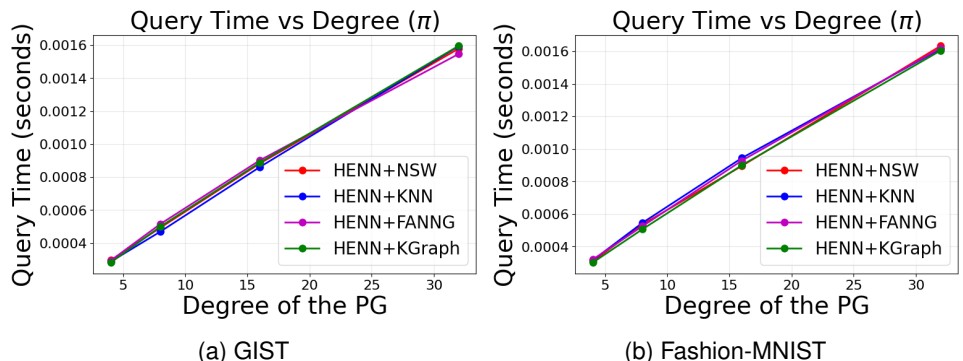

Figure 20: Effect of the degree of PG on the query time. Every other parameter, like the number of layers, is fixed.

In contrast, NSW is able to escape these local traps more effectively, resulting in a distribution that concentrates more heavily on smaller values of $K$.

**Distribution of** $T$. Figure 23 shows the derived distribution of the running-time variable $T$ using equation 3. Here, we choose an arbitrary constant $c$, so the resulting values do not correspond directly to the exact number of visited hops; rather, they represent a scaled version of the running time. The goal is to enable a meaningful *relative* comparison across different proximity graphs and dataset settings, which this transformation preserves.

**Expected Value of** $T$. Figure 24 reports the expected value of $T$ across different proximity graphs and datasets, computed using the result of Theorem 4 and equation 3. These values reflect the predicted average running time of HENN under each proximity graph construction (again, here $T$ is proportional to time).

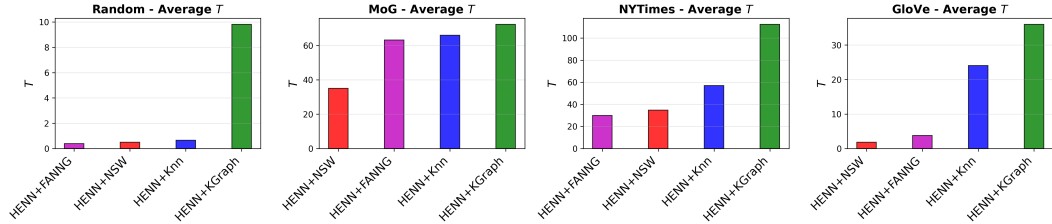

Figure 24: Expected values of $T$.

**Real Query Time of HENN.** Figure 25 presents the actual query-time performance of HENN constructed using different proximity graphs. The empirical runtimes exhibit patterns consistent with

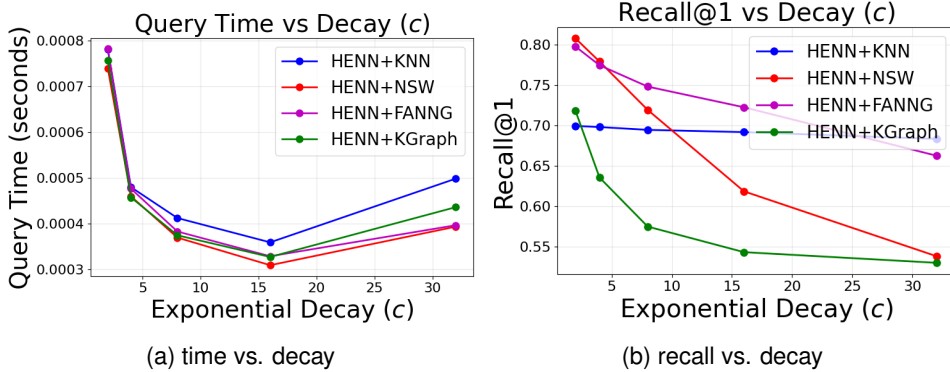

(a) time vs. decay    (b) recall vs. decay

Figure 21: Effect of exponential decay rate on time and recall of HENN. Higher decay rate is equivalent of smaller number of layers. The experiments are on synthetic dataset with Mixture of Gaussians with 10k points and $d = 4$.

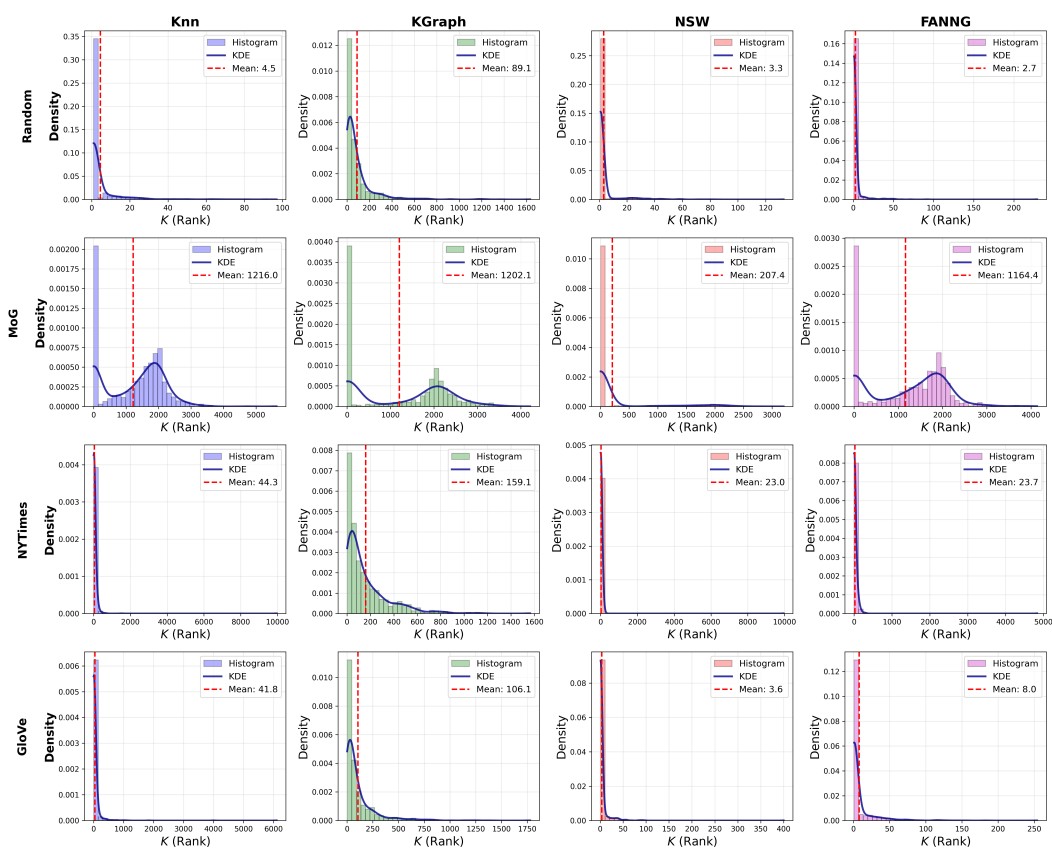

Figure 22: Distribution of $K$.

our theoretical predictions: the relative ordering and average behavior closely mirror the expected values of $T$, thereby validating our analysis.

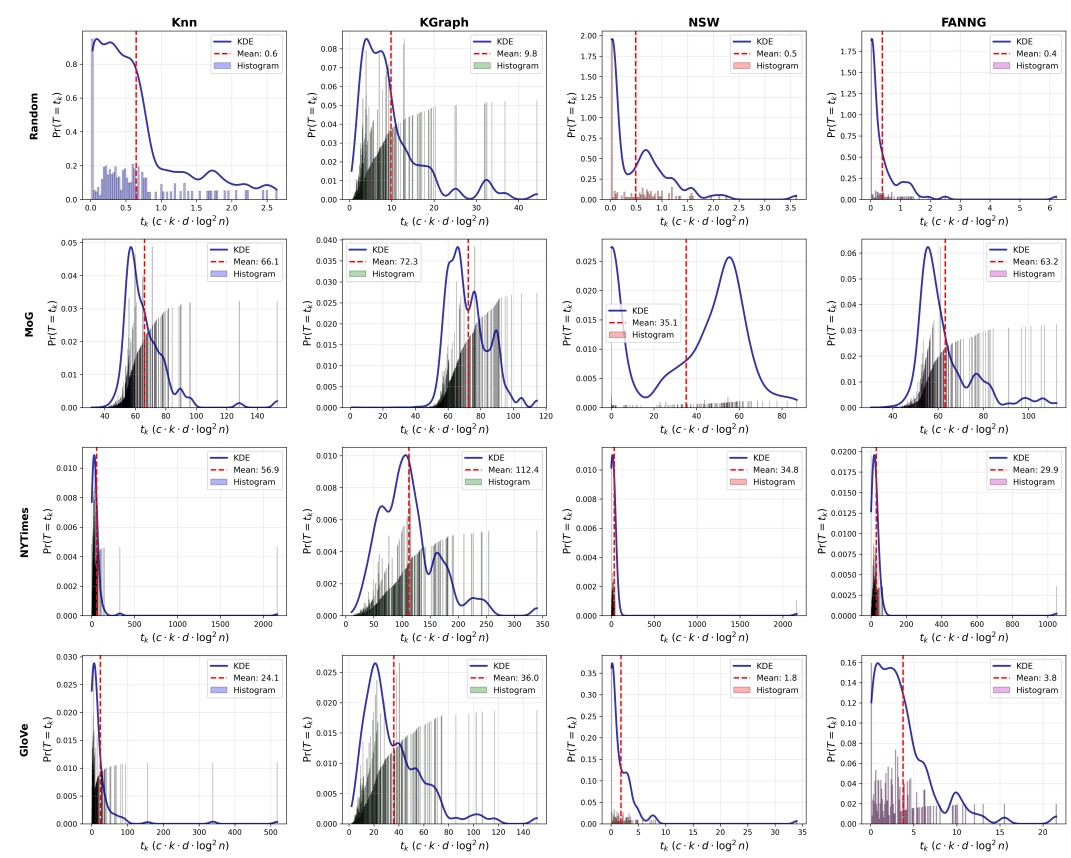

Figure 23: Distribution of $T$.

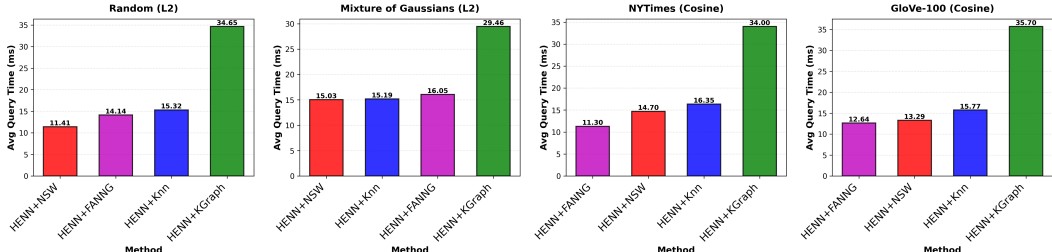

Figure 25: Real observed average query times of HENN.

