# OpenReview forum: "Hierarchical Epsilon-Net Graphs: Time Guarantees for HNSW in Approximate Nearest Neighbor Search"
_ICLR.cc/2026/Conference — ICLR 2026 Conference Withdrawn Submission_

### Official Review · Reviewer_zSAp · 2025-10-29

**Soundness:** 4
**Presentation:** 4
**Contribution:** 2
**Rating:** 4
**Confidence:** 4

**Summary:**

This paper introduces _Hierarchical $\epsilon$-net Navigation_ (HENN), a property for hierarchical graph data structures such as the ever-popular HNSW. This property revolves around $\epsilon$-nets, a common tool in computational geometry. For a HENN graph, one can think of each layer in a hierarchical data structure as an $\epsilon$-net of the larger prior layer. We connect the hierarchy by drawing edges between common points across layers. Producing this data structure thus gives provable guarantees on HNSW:
1. Establishing that HNSW is a HENN graph with high probability
2. HNSW thus has logarithmic query times, conditioned on a recall bound parametrized by a success probability $\gamma$

The authors also provide a practice-friendly algorithm to compute a HENN graph, which offers a tradeoff between speed and quality (which existing algorithms cannot give). Finally, the authors provide empirical comparisons on standard retrieval datasets, such as SIFT, NYTimes, and GIST. The results are meant to be complementary to the theoretical results and show (nearly) matching performance between HNSW and HENN + NSW.

**Strengths:**

1. Theoretical understanding of graph-based ANNS is lacking and this submission contributes a very necessary result in that regard.
2. The main body is well-written, concise, and easy to follow.
3. The idea of applying $\epsilon$-nets to model graph-based ANNS is novel. I particularly like that it provides a model for multiple graph approaches.

**Weaknesses:**

1. The budget algorithm for computing $\epsilon$-nets is an interesting contribution, but it seems hard to justify using it in practice when we could just use the well-optimized HNSW. Not to mention, it's still probabilistic.
2. As an extension, I think the empirical results feel like an afterthought. My takeaway from this paper was that we could get theoretical guarantees for HNSW and other hierarchical graphs using $\epsilon$-nets, but it would've been nice to see a wider empirical study of HENN to complement it. For example, I'd have liked to have seen results on how HENN scales with dataset size and a series of plots tracking the recall bound (or an appropriate surrogate) with respect to QPS.
3. I think a larger discussion about the place of this result among other recent works in theoretical graph-based search would be useful, simply to compare approaches and to help the reader understand the merits of this paper's approach.
4. As this is primarily a theoretical submission, I don't know if I agree with the discussion in appendix A: I believe some kind of some kind of worst-case analysis on the output quality of HENN/HNSW would have been nice. I think such a result would've made for a more complete submission.

**Questions:**

1. How difficult would it be to extend the results of this work to a beam search setting? i.e. can we obtain provable guarantees for outputting k > 1 nearest neighbors?

---

> ### Author Response · Authors · 2025-11-21
>
> We thank the reviewer for the comments. We address them individually in the following.
>
> **Weakness 1: The budget algorithms...**
>
> The budget-aware algorithm introduces a trade-off between the probability that the final subset forms an $\varepsilon$-net and the preprocessing time spent constructing it. This trade-off allows us to obtain a **more compressed graph** when desired (Figure 6). HNSW corresponds to the case where we do not spend additional preprocessing time beyond what HNSW already does, but if one is willing to invest more preprocessing time, the budget-aware algorithm can yield significantly more compact layers, as shown in Figure~6.
>
> **Weakness 2: As an extension...**
>
> The main contribution of this work is defining a general class of hierarchical graph indices (HENN) and, as a result, providing probabilistic running-time guarantees for HNSW, something that was not available before. Thank you for the suggestion; in the camera-ready version we will include additional experiments, including scaling behavior on larger datasets from the Big ANN Benchmark and plots examining how HENN scales with dataset size as well as how the recall bound of the proximity graph correlates with QPS.
>
> **Weakness 3: I think a larger discussion...**
>
> Thank you for the comment. In our revised PDF version, we added a more detailed discussion situating our results within recent theoretical work on graph-based search, to better highlight the relationship between our approach and other frameworks in the literature. We edited the related work section also based on the suggestions of other reviewers.
>
> **Weakness 4: As this is primarily...**
>
> Thank you for the suggestion. As highlighted in the literature, carefully constructed adversarial examples can force any greedy graph-based ANN method into linear worst-case behavior. We will incorporate additional experiments examining the worst-case behavior of HENN in the camera-ready version.
>
> **Question 1: How difficult would it be to extend...**
> This is a good question. In beam search, a candidate set of size $C$ is maintained at each step, and multiple nodes are expanded instead of just one. Our initial observation is that the complexity should scale by roughly a factor of $C$, meaning that the running-time bounds would extend to beam search by multiplying the corresponding terms by~$C$.

---

### Official Review · Reviewer_MBvW · 2025-10-30

**Soundness:** 2
**Presentation:** 3
**Contribution:** 2
**Rating:** 2
**Confidence:** 4

**Summary:**

The authors introduce a property of hierarchical graphs called Hierarchical ε-Net Navigation (HENN) and analyze the time bounds for ANN search on graphs that satisfy the HENN property. The authors also claim that HNSW satisfies the HENN property with high probability and derive time guarantees for HNSW. The authors also conduct the experiments to show the effectiveness of HENN.

**Strengths:**

S1: The theoretical analysis of HENN is appreciated.

S2: The experimental evaluation of HENN is extensive.

**Weaknesses:**

W1: The theoretical analysis seems not very robust. The parameter $\rho_{\gamma}$ plays a key role in the theoretical results, and the authors claim that, for most existing navigable graphs, $\rho_{0.9} = O(1)$ (line 252). However, this claim is not rigorous. In fact, for some hard datasets, the recall of HNSW cannot even reach 0.9. Therefore, I wonder whether the theoretical analysis can truly explain why HNSW performs well in practice.

W2: I could not find an ablation study. The authors test several similarity graphs equipped with HENN, but I would like to see a direct performance comparison between using HENN and not using HENN.

W3. From the results in Figure 7, HENN+NSW does not show any performance improvement over HNSW. Therefore, I wonder whether it is meaningful to use HENN in practice.

W4: There exist several other state-of-the-art ANNS methods [a][b][c] that are not discussed in the related work. The authors are encouraged to discuss and compare their method with these approaches. For readers, it would be more informative to see a comparison between HENN-based similarity graphs and these SOTA approaches, as the hierarchical structure is not indispensable for many SOTA ANNS solvers.

[a]. Accelerating large-scale inference with anisotropic vector quantization. ICML 2020

[b]. Probabilistic routing for graph-based approximate nearest neighbor search. ICML 2024

[c]. Rabitq: Quantizing high-dimensional vectors with a theoretical error bound for approximate nearest neighbor search. SIGMOD 2024

**Questions:**

My comments and suggestions are as follows:

C1: Could the authors provide a more rigorous analysis of $\rho_{\gamma}$ . The current explanation is not convincing. (See W1)

C2: The ablation study is important, and the authors are encouraged to include the ablation experiment mentioned in W2.

C3: Could the authors clarify the practical contribution of this paper? (See W3)

C4: The discussion of related work should be expanded to provide a more comprehensive comparison with existing studies so that readers can better understand the position of this paper. (See W4)

---

> ### Author Response · Authors · 2025-11-21
>
> We thank the reviewer. We believe there was a misunderstanding regarding our contribution. Our main contribution is to define a general class of hierarchical graph-based indices for ANN and to provide a probabilistic time bound for HNSW. Below, we address the concerns individually.
>
> **W1:**
>
> Thank you for pointing this out. In the revised PDF version, we have added a new appendix section titled *“More on Recall Bound’’*. Please also see our response to reviewer "EaJH". Our analysis is not limited to cases where $\rho_\gamma = O(1)$. Whatever the value of $\rho_\gamma$ is for a given proximity graph directly appears in the running time, which is $O(\rho_\gamma d \log^2 n)$.
>
> We fixed $\gamma$ in the theorem only to simplify the presentation. As discussed in the new appendix section, by varying $\gamma$ one obtains a full probability distribution over the running time of HENN. Thus, our analysis does not rely on the assumption that $\rho_\gamma$ is constant; the running time scales proportionally with its actual value.
>
> Please refer to our revised PDF version where this misunderstanding is resolved by rewriting the paragraph before Section 4.2 and adding a new Appendix section to further clarify this. We also changed the Theorem 4 statement and include the dependence on $\rho_\gamma$ and $\pi$ in the final bound.
>
> **W2 and W3:**
>
> We believe there may be a misunderstanding. Our goal is not to introduce a new graph index intended to outperform existing ones. Instead, our contribution is to provide *probabilistic running-time guarantees* for HNSW by introducing a general class of graphs called HENN. Figure~7 is included specifically to demonstrate that “HNSW is a HENN graph’’, that is, HNSW is structurally a special case of HENN, and therefore our theoretical bounds on HENN automatically apply to HNSW. Hence, the purpose is not to show empirical gains of HENN over HNSW, but to show that our theory correctly matches HNSW’s behavior.
>
> **W4:**
>
> Thank you for mentioning these works. Our focus is on graph-based ANNS algorithms, and our theoretical framework is specific to this family. Methods such as Rabitq and the approaches in [a] and [c] are primarily *quantization-based* or use techniques outside the graph-based paradigm, and therefore are not directly comparable to our analysis.
>
> In the revised PDF version we expanded the related work section and included your suggested literature. We also clarified the distinction between graph-based methods and other categories of ANN solvers. Please see the last two paragraphs added in "Other solutions for ANN" subsection in the related work.

---

> > ### Comment · Reviewer_MBvW · 2025-11-28
> >
> > Thank you for your explanation. I have read the new appendix section, but I have to say that I still do not see strong theoretical insights in this part. The relationship between $\gamma$ and $\rho_{\gamma}$ appears to depend heavily on the specific graph structure and is hard to predict. What is the worst case? How do we know that $\rho_{\gamma}$ is not very large in practice? I feel the core theoretical issue is exactly this relationship. Since the authors position this work as a theoretical paper, I feel that the current contributions remain somewhat limited. On the other hand, many other graph structures, such as DiskANN and NSG, do not use a hierarchical structure but still achieve even better search performance, which suggests that the hierarchical structure might not be the key reason why similarity graphs perform very well in practice.

---

> > > ### Author Response · Authors · 2025-11-28
> > >
> > > Thank you for carefully reading our response and the revised paper, and for your thoughtful comment. We agree that the reported running time depends on the specific proximity graph used at each layer.
> > >
> > > ### *About the Theoretical Contribution*
> > >
> > > Our work is best interpreted as a general framework for hierarchical graph–based indexes. One of our main contributions is providing the (probabilistic) time bound for HNSW, an index that is widely used across all modern vector database systems, yet previously lacked formal guarantees. We hope this line of work explores the theoretical understanding of such structures and, more broadly, of any ANN graph-based index built **based on sampling**. Viewing these constructions through the lens of $\varepsilon$-nets provides a new perspective that we believe will be valuable.
> > >
> > > We would also like to highlight that this work generalizes to any black-box proximity graph given by the user. We believe this level of generality is important. While it is difficult to provide full theoretical guarantees for every existing proximity graph, our approach formalizes a notion of performance (a recall-style bound) that is sufficient for deriving end-to-end runtime guarantees for the resulting hierarchical index.
> > >
> > > We would also like to emphasize that this interpretation of the HNSW index enables additional structural improvements. Viewing each layer as an ε-net allows us to further **compress the index** by pruning unnecessary nodes (as illustrated in Figure 6). Since the objective is ultimately to obtain an $\varepsilon$-net, the same guarantees can be achieved with fewer points if additional preprocessing time is available. In contrast, simple random sampling is not space-efficient and results in **redundancy**.
> > >
> > > We also agree that there exist other proximity graphs that provide strong empirical performance and meaningful guarantees. We have included NSG and several of the alternatives in our experiments and compared them against other commonly used proximity graphs. However, providing a comprehensive discussion of all existing proximity graph variants is challenging, given the large number of designs and differences among them. Our goal in the paper is therefore to present a general framework on a broad class of such graphs rather than cataloging every individual variant.
> > >
> > > ### **About Flat Indices**
> > >
> > > Regarding **flat indices**, we agree that they are important, and we don’t deny that. While our main focus is hierarchical, sampling-based indices (such as HNSW), our framework naturally generalizes to flat indices. As explained in our response to reviewer hbAq, this generalization works as follows:
> > >
> > > 1. One can use $\varepsilon$-net samples to extract a well-spread subset of points
> > > from a standard flat graph index. $\varepsilon$-net sampling selects nodes that are spatially well-distributed across the dataset.
> > >
> > > 2. By connecting these well-spread nodes, we add **shortcut edges** that accelerate search. Unlike heuristic shortcut-insertion methods, our approach is provably correct: Theorem 4 shows that after taking one of these shortcuts, the remaining number of steps is bounded, due to ε-net properties.
> > >
> > > 3. The $\varepsilon$-net viewpoint also enables **compression**. As discussed in Figure 6: ε-net samples capture the “essential” points needed for good performance. With more preprocessing time, one can compute ε-net samples using fewer points, giving a more compressed index. Thus, the same **preprocessing vs. compression trade-off** extends to flat indices.
> > >
> > > We thank the reviewer for the comments. We will add a clearer discussion explaining how our framework generalizes to flat indices. This direction also opens an interesting path for future work, potentially leading to a new flat index, a more compressed variant of NSW, derived directly from our ε-net–based interpretation.
> > >
> > > In summary, we respectfully disagree that our contribution is limited. Beyond the theoretical analysis, our work also develops **practical heuristic algorithms** for constructing such indexes (and ε-nets), that to the best of our knowledge, were not available in the literature and are directly usable for real index constructions. In addition, we provide a comprehensive empirical study showing how different proximity graphs behave within hierarchical indices in practice.
> > >
> > > Altogether, our work offers a deeper understanding of how sampling-based spatial graph indices operate on point sets: what random sampling represents from a computational geometry perspective, how diverse and well-spread the selected points are, how these indices behave probabilistically on general (instance-agnostic) data, and why HNSW exhibits logarithmic performance on most datasets in practice.

---

### Official Review · Reviewer_ii1u · 2025-10-30

**Soundness:** 4
**Presentation:** 4
**Contribution:** 3
**Rating:** 6
**Confidence:** 5

**Summary:**

The paper introduces a graph property called Hierarchical epsilon Net Navigation(HENN). They provide probabilistic query-time bounds for the graphs that satisfy HENN property, that are logarithmic in input parameters. Furthermore, they show that the famous and widely used HNSW graphs satisfy HENN property with high probability. Finally, they design a novel budget-aware algorithm that with more preprocessing time increases the probability of successfully constructing an epsilon-nets, which is a key-subroutine to construct HENN graphs.


Decision: I feel the paper should be accepted.

**Strengths:**

1) Paper is well very written and clean. It is easily readable, and quite fun to read. Proofs are well written, loved reading the paper in general.

2) The paper shows nice theoretical guarantees for HENN graphs, which covers popularly and widely used HNSW.

3) Building relation between epsilon nets and recall bound is nice.

4) Connection between HENN and HNSW is nice.

5) Figure 6, showing reduction of index size with higher processing time, is extremely nice. This is happening because with higher processing time one is able to construct smaller sized epsilon nets.

**Weaknesses:**

1) The guarantees are in terms of recall bounds definition which is non-traditional.

2) Given the definition of recall bound, I felt the proofs are simple.

3) The success probability is exponential in log n, that is gamma^{log n}, and gamma is something like 0.9 in practice for constant recall bound. Together this implies that success probability is very tiny.

**Questions:**

1) Please state the running time in terms of the maximum degree of the proximity graph and the recall bound parameter.

2) Can you please elaborate a more on "For most existing navigable graphs, Malkovetal.[36]; Malkov&Yashunin[35] show
 that ρ_{0.9}=O(1). Particularly, is this a theoretical result or just emprical finding? Would you please also mention the degree of these proximity graphs. In particular, I wish to know if we know of proximity graphs, that achieve constant recall bound with constant degree proximity graphs.

3) Index size: I believe roughly captures the number of edges? It would be interesting to see an analogous to Figure 6, where the instead of index size, you show the reduction in just the number of nodes.

4) What are practical implications of these theoretical findings? One is reduction in index size, by using your budget-aware procedure, is there anything else? You can be very speculative here.

5) Also, when you define recall bound, you need to specify the definition of distribution of choice for start note s. For theorem 4 to hold, what would this distribution be. I feel you may want to change the definition of recall bound where instead of doing a random choice s, you write recall bound over worst case start node s, which will change definition as follows: min{k|\max_{s} Pr_{q}[GS_{q}(G,s) \in NN_{k,X}(q)] >= \gamma}. Without this, your distribution of s, will need to depend on the query q itself. Happy to elaborate further if this doesn't make sense.

6) In Theorem 4, can you also write the expression for recall bound of HENN graph in terms of the recall bound of PG at each level.

7) Would you please talk more about the success probability which is exponential in log n. In particular, why is this okay?

---

> ### Author Response · Authors · 2025-11-21
>
> We would like to thank the reviewer for the very insightful comments. Below, we address the questions.
>
> **Question 1: Please state the running time in terms of...**
>
> The running time depends proportionally on both the maximum *degree of the proximity graph* and the *recall bound*. If $\pi$ is the maximum degree, then the running time is: $O(\pi \rho_\gamma d \log^2 n)$.
>
> Further clarification on how the recall bound is defined and used appears in our response to reviewer “EaJH’’ and in the new appendix section (revised PDF version) titled “More on Recall Bound.’’
>
> In the revised PDF, we also changed the statement to also include the degree $\pi$ and recall bound $\rho_\gamma$ in the time.
>
> **Question 2: Can you please elaborate...**
>
> Please also see our response to reviewer “EaJH.’’ In the revised version, we added a new appendix section titled “More on Recall Bound’’ that explains this more clearly. In the theorem statements we fixed a value of $\gamma$ only to simplify presentation, but in general we do not rely on a fixed recall bound. By varying $\gamma$, one can obtain a full probability distribution of the final running time of a HENN graph (as previously explained in the paragraph after Theorem 4 and before Section 5 and also explained in the new added appendix section). Thus our result should be interpreted as providing a probabilistic distribution over running time, not a single fixed value. In general, if for a proximity graph the value of recall bound is large, this result in the longer running time in the corresponding HENN index.
>
> We changed the paragraph before Section 4.2 that caused this misunderstanding and referred to the Appendix section for further details.
>
> **Question 3: Index Size...**
>
> The figures for index size capture both nodes and edges. However, in our setting the degree is fixed, meaning the number of edges is almost the same across methods (HNSW, HENN+NSW, etc). Therefore, the main source of variation comes from the number of nodes per layer, which is exactly what the budget-aware algorithm controls. We are not changing the edge-selection logic. Thank you for the suggestion; we will add a new figure showing the reduction in the number of nodes in the camera-ready version.
>
> **Question 4: What are practical implications...**
>
> The budget-aware procedure provides a trade-off between the probability of finding an $\varepsilon$-net and the amount of preprocessing time. To obtain a more **compressed index**, one can spend additional preprocessing time as indicated in our analysis (Figure 6). More broadly, building $\varepsilon$-nets is useful in many areas (e.g., database systems), so a budget-aware approach is independently valuable. Our other main practical implication is providing probabilistic running-time guarantees for hierarchical graphs such as HNSW, which is an active topic of research.
>
> **Question 5: Also, when you define...**
>
> Thanks for raising this point. In our definition, the probability is taken over all possible choices of the starting node $s$, depending on how the PG selects it. Some proximity graphs choose $s$ randomly, while others deterministically choose a fixed entry point. Your suggestion of defining the recall bound using the worst-case starting node also makes sense and is compatible with our analysis. Theorem~4 would still hold, since it provides an upper bound. We will clarify this in the revised definition.
>
> **Question 6: In Theorem 4, can you also...**
>
> This is a very good question. Please also refer to the discussion in the new appendix section “More on Recall Bound.’’
> If the recall bound uses a random starting point, then after descending through the hierarchy, the point in the last layer is typically much closer to the query. Therefore, its recall distribution is no longer the same as the original PG, and we may not directly derive a simple closed-form *recall bound for HENN* itself. However, based on the random variable $K$ defined in the new appendix section, focusing on a smaller region should logically make the distribution of $K$ more concentrated on neighbors near $q$, increasing the probability of returning a closer point.
> In the second case, using your suggested worst-case start node, the recall bound of HENN becomes the same as the recall bound of the base PG. We will clarify both interpretations in the camera-ready version.
>
> **Question 7: Would you please talk more about...**
>
> This probability corresponds to obtaining the runtime $O(\rho_\gamma \, d \log^2 n)$ when $\gamma$ is fixed to a particular value. However, as discussed in the appendix section “More on Recall Bound,’’ this expression is only for a fixed $\gamma$. By viewing the returned rank $K$ as a random variable, varying $\gamma$ yields a full distribution of the final runtime of HENN. Thus the exponential-in-$\log n$ success probability is part of the derivation for a fixed parameter, while the full theory describes a probability distribution over the running time.

---

### Official Review · Reviewer_hbAq · 2025-11-01

**Soundness:** 3
**Presentation:** 3
**Contribution:** 2
**Rating:** 4
**Confidence:** 4

**Summary:**

This paper studies the problem of developing theoretical guarantees on the latency and search quality of hierarchical graph-based approximate near-neighbor search algorithms like HNSW. This is a very important and timely research question since these algorithms have achieved widespread adoption and strong performance in practice, but still largely lack meaningful theoretical guarantees. This paper makes progress in closing this gap between theory and practice by drawing a connection between computational geometry and learning theory. In particular, the authors introduction the notion of hierarchical epsilon-net navigation graphs (HENN), show that HNSW is an instantiation of HENN, and utilize this framework to prove a probabilistic guarantee on the runtime of HNSW, which they show is poly-logarithmic in the number of hierarchical layers and the vector dimensionality with high probability. The authors also experimentally validate their theorems where they find strong empirical evidence in support of their theoretical claims.

**Strengths:**

This paper makes progress on an important research problem in providing principled theoretical guarantees on the query complexity of graph-based near neighbor search algorithms like HNSW. Moreover, the authors experimentally validate their theory which strengthens their claims considerably. The paper is also well-written with clear plots, figures, and theorem statements while deferring the appropriate amount of detail to the appendix. The core insight at the heart of the paper in connecting ideas in computational geometry, namely epsilon-nets to concepts in learning theory such as VC dimension is creative and may inspire future work within this direction as well. The authors also use their theory to design an improved indexing scheme that reduces memory by spending more time preprocessing. This is also a novel insight and may have practical implications as well.

**Weaknesses:**

1. I think the experimental section is currently weak in that it does not consider large scale datasets beyond roughly 1 million points. I would strongly encourage the authors to consider running experiments on datasets such as Big ANN Benchmarks, which include benchmarks at the 10M, 100M, and 1B scale. I believe that running experiments at scale is especially important for this work because the authors claim that the hierarchical layers improve scalability, but this claim is difficult to verify without large-scale validation.

2. The current discussion of related work is too brief and should not be completely relegated to the appendix because acknowledging the relevant pieces of related work is important for understanding the contributions of the paper. In particular, the authors discuss related work on hierarchical graph-based ANN algorithms, but do not really mention any non-hierarchical graph-based techniques that are also achieve state-of-the-art performance, such as [Vamana](https://papers.nips.cc/paper_files/paper/2019/file/09853c7fb1d3f8ee67a61b6bf4a7f8e6-Paper.pdf). In addition, multiple works in the literature seem to have recently demonstrated the hierarchical layers of HNSW are not necessary ([Lin & Zhao, 2019](https://arxiv.org/pdf/1904.02077), [Coleman, et al, 2022](https://arxiv.org/pdf/2104.03221), and [Munyampirwa, et al. 2025](https://openreview.net/pdf?id=OJwITuuU3h). This large body of literature on non-hierarchical graph-based ANN likely merits some discussion in this paper, and the authors might also want to address whether their theorems are consistent with these results as well.

3. The citations in the paper are currently in the wrong format. The authors should follow the ICLR guidelines and use the natbib package and apply citations with the \citep{} and \citet{} commands as appropriate.

**Questions:**

1. As mentioned above, I would strongly suggest that the authors consider experimentally validating the core claims of the paper on larger-scale benchmark datasets, such as those from Big ANN benchmarks since scalability is a core component of the narrative in the paper.

2. How does Theorem 4 of the paper change when the number of layers is 1? Is Theorem 4 consistent with the growing body of research in the literature suggesting that hierarchical layers are not required for state-of-the-art performance in graph-based near neighbor search (particularly in the context of [Vamana](https://papers.nips.cc/paper_files/paper/2019/file/09853c7fb1d3f8ee67a61b6bf4a7f8e6-Paper.pdf) and the findings of [Munyampirwa, et al. 2025](https://openreview.net/pdf?id=OJwITuuU3h). Can the theory in the paper be naturally extended to handle the case of a single layer graph?

---

> ### Author Response · Authors · 2025-11-21
>
> We thank the reviewer for the feedback. We did address the concern regarding related work in the revised PDF version by discussing the suggested papers. Below, we address the concerns individually.
>
> **Weakness 1 and Question 1: I think the experimental section is currently...**
>
> We appreciate this important suggestion. We agree that validating scalability at larger scales is essential. We will include experiments on the Big ANN Benchmarks in the camera ready version.
>
> **Weakness 2: The current discussion of the related work...**
>
> Thanks for bringing these related works to our attention. In our revised PDF version, we updated the related work section and included the suggested literature. Please look at the last two paragraphs of the related work in Appendix A. The related literature discuss that the main advantage of HNSW is the existence of hubs. Our result can also be translated to the time bounds when there are hubs (shortcutts) in the graph while the greedy algorithm proceeds.
>
> **Question 2: How does Theorem 4 of the paper...**
>
> That's a good discussion, thank you for bringing it up. If there is only one layer, one can use a similar $\varepsilon$-net sampling idea to modify the edges (long-link connections) between different regions of the space in the graph. In particular, a long edge connecting two nodes of an $\varepsilon$-net (a subset of vertices) can be added to a single flat graph to enable **shortcutting**. In this way, our theorem can still bound the **number of additional steps** required (after taking such a shortcut) to reach the final answer.

---

### Official Review · Reviewer_EaJH · 2025-11-03

**Soundness:** 2
**Presentation:** 4
**Contribution:** 2
**Rating:** 2
**Confidence:** 4

**Summary:**

This paper provides a theoretical analysis of the running time and search quality of the Hierarchical Navigable Small World (HNSW) algorithm.
The main idea is that, with high probability, the randomly sampled upper layers in HNSW form an $\varepsilon$-net of their corresponding lower layers.
Under the "navigable graph assumption", the search result from the upper layer is already an approximate nearest neighbor for the next layer, allowing the algorithm to refine results through the hierarchy.
By analyzing the running time, the authors argue that the optimal hierarchical structure reduces data size by a constant factor c per layer, leading to a total of $O(\log n)$ layers and a query time of $O(d \log^2 n)$
The paper also includes empirical evaluations demonstrating that the theoretical predictions align with practical observations on HNSW and other popular navigable graph structures.

**Strengths:**

1. The theoretical analysis of HNSW’s performance addresses a key open question given the algorithm’s widespread use in large-scale ANN search.
2. The authors present both mathematical analysis and comprehensive experiments, with proofs that are relatively accessible and experiments that are detailed.
3. The use of $\varepsilon$-net to connect the optimality of the approximate neighbors across hierarchical layers is an elegant idea.

**Weaknesses:**

1. The theoretical results depend heavily on the assumption that $\rho_{\gamma}$ (the recall bound of a navigable graph) is a constant and that greedy search on a navigable graph with constant degree reliably returns a top-$\rho_{\gamma}$ neighbor. While this assumption may hold empirically, it fails in worst-case scenarios (as shown by Dian et al. [9]), which limits the theoretical soundness of the claimed guarantees for HNSW.
2. Even if the greedy search can consistently find a top $\rho_{\gamma}$ neighbor, the overall success rate of the hierarchical search decays exponentially with the number of layers—approximately $\gamma^{\log n}$. Thus, even with $\gamma \approx 0.99$, the success rate tends toward zero as $n$ grows...

**Questions:**

1. Could the authors clarify Algorithm 4 $\varepsilon$-net construction, particularly the step FindUnhitRange?
    It is stated to run in $O(n)$ time (around line 869), but I'm still unclear about the details.
2. Minor typo: $L_L$ on line 1108

---

> ### Author Response · Authors · 2025-11-21
>
> We appreciate the reviewer’s constructive feedback. We note that part of the confusion comes from a misunderstanding of the recall bound, which we now clarify in a newly added appendix section K titled “More on Recall Bound.” Please see the revised PDF. We then address the reviewer’s comments one by one below.
>
> **Weakness 1: The theoretical results depend heavily on ...**
>
> We would like to clarify a misunderstanding regarding the role of $\rho_\gamma$. Our analysis *does not assume that $\rho_\gamma$ is constant*. The final bound scales as $O(\rho_\gamma \cdot d \cdot \log^2 n)$,
> so if a proximity graph has a large recall bound, the running time increases proportionally (as mentioned in the paragraph after Theorem 4 statement before Section 5). We **fixed** $\rho_\gamma$ only to simplify the exposition and to highlight dependencies on $n$ and $d$. Thus, if a proximity graph has a poor recall bound, our theory accurately reflects that it will perform poorly; we do not rely on $\rho_\gamma$ being constant.
>
> To further clarify this, we added a new appendix section in the revised PDF, titled *``More on Recall Bound''*. This section formally explains the recall bound, how it should be interpreted, and how our theoretical results map the empirical distribution of returned ranks $K$ to a full distribution of running times $T$. A high-quality proximity graph produces a distribution concentrated on small $T$, which we validate with additional experiments included in that appendix section.
>
> Finally, we emphasize that the goal of our work is to provide *probabilistic* running-time guarantees for HENN (and HNSW-style) graph indices. While worst-case failures are known (e.g., Dian et al.~[9]), we are not providing worst-case upper bounds. Instead, our analysis characterizes probabilistic behavior under any given settings.
>
> In order to prevent this misunderstanding, in the revised PDF, we changed the final paragraph (before section 4.2) and discussed what *``fixing the recall bound''* means. Also we changed the Theorem 4 statement and provided the full final time that depends on $\rho_\gamma$.
>
> **Weakness 2: Even if the greedy search...**
>
> This concern is also addressed in the new appendix section *``More on Recall Bound''*. The expression $\gamma^{\log n}$ appears only when fixing $\gamma$ in the derivation of Theorem~4. In general, we do **not** fix $\gamma$; rather, we consider the full empirical distribution of the returned rank $K$ (and the induced $\rho_\gamma$ values across different $\gamma$). Our theorem then maps this empirical distribution to the resulting distribution of the running time $T$. Therefore, $\gamma^{\log n}$ is not the overall success probability but an intermediate expression for a fixed parameter. Varying $\gamma$ yields the full runtime distribution predicted by our theorem.
>
> **Question 1: Could the authors clarify...**
>
> A naïve implementation of `FindUnhitRange` would require checking all possible ranges, which is $O(n^d)$. Instead, we use the following heuristic: we partition the space into a constant number of equally sized rings (constructed over the full dataset) and check only these $O(1)$ partitions to see whether any do not intersect the current $\varepsilon$-net. If so, that partition is returned as an unhit range. Although this heuristic does not give a formal worst-case guarantee, our experiments show that we can almost always find unhit ranges this way. As a result, as long as one can find one unhit range at each iteration, the algorithm can proceed.
>
> **Question 2: Minor typo line 1108**
>
> We thank the reviewer for pointing this out. However, $\mathcal{L}_L$ **is not a typo** here. It corresponds to the highest layer, where $L$ is the number of layers. $\mathcal{L}_1$ being the first layer, and then $\mathcal{L}_2$, and so on.

---

### Author Response · Authors · 2025-11-21

We thank all reviewers for their valuable feedback. We emphasize that the main contribution of our paper is threefold: (1) defining a general class of hierarchical graph-based ANN indices, (2) establishing the first probabilistic running-time guarantees for HNSW-style structures, and (3) introducing the first budget-aware algorithm for constructing $\varepsilon$-nets, also yielding a more compressed index.

Several concerns stemmed from misunderstandings of the recall bound and the interpretation of Theorem 4; these are now fully clarified in the newly added Appendix K: “More on Recall Bound,” together with additional empirical validation (in the new revised PDF).

---

### Note · Authors · 2026-01-06

I have read and agree with the venue's withdrawal policy on behalf of myself and my co-authors.